# Deletion of FNDC5/irisin modifies murine osteocyte function in a sex-specific manner

**Anika Shimonty[1], Fabrizio Pin[1], Matthew Prideaux[1], Gang Peng[1], Joshua Huot[1], Hyeonwoo Kim[2], Clifford J Rosen[3], Bruce M Spiegelman[4], Lynda F Bonewald[1,5]\***

[1]Indiana University, Indianapolis, United States; [2]Korea Advanced Institute of Science and Technology, Daejon, Republic of Korea; [3]Maine Health Access Foundation, Portland, United States; [4]Dana Farber Cancer Institute, Boston, United States; [5]Indiana Center for Musculoskeletal Health, Indianapolis, United States

**\*For correspondence:** lbonewal@iu.edu

**Competing interest:** The authors declare that no competing interests exist.

**Preprint posted** 06 November 2023

**Sent for Review** 06 November 2023

**Reviewed preprint posted** 27 December 2023

**Reviewed preprint revised** 13 March 2024

**Version of Record published** 25 April 2024

**Abstract** Irisin, released from exercised muscle, has been shown to have beneficial effects on numerous tissues but its effects on bone are unclear. We found significant sex and genotype differences in bone from wildtype (WT) mice compared to mice lacking *Fndc5* (knockout [KO]), with and without calcium deficiency. Despite their bone being indistinguishable from WT females, KO female mice were partially protected from osteocytic osteolysis and osteoclastic bone resorption when allowed to lactate or when placed on a low-calcium diet. Male KO mice have more but weaker bone compared to WT males, and when challenged with a low-calcium diet lost more bone than WT males. To begin to understand responsible molecular mechanisms, osteocyte transcriptomics was performed. Osteocytes from WT females had greater expression of genes associated with osteocytic osteolysis and osteoclastic bone resorption compared to WT males which had greater expression of genes associated with steroid and fatty acid metabolism. Few differences were observed between female KO and WT osteocytes, but with a low-calcium diet, the KO females had lower expression of genes responsible for osteocytic osteolysis and osteoclastic resorption than the WT females. Male KO osteocytes had lower expression of genes associated with steroid and fatty acid metabolism, but higher expression of genes associated with bone resorption compared to male WT. In conclusion, irisin plays a critical role in the development of the male but not the female skeleton and protects male but not female bone from calcium deficiency. We propose irisin ensures the survival of offspring by targeting the osteocyte to provide calcium in lactating females, a novel function for this myokine.

## eLife assessment

The study presents **valuable** findings on sexually dimorphic patterns of osteocytic transcriptomes and low calcium diet-induced osteocytic osteolysis in FNDC5-deficient mice. The authors present **solid** evidence for sex-specific changes in osteocyte morphology and gene expression under a calcium-demanding setting in this particular strain of mice, although the protective role of FNDC5-deficiency in lactation and low-calcium diet in female mice remains unclear due to lack of mechanistic studies. The study also lacks evidence that irisin, a proteolytically cleaved product of FNDC5, is responsible for the observed phenotypes, as irisin was not directly measured.

## Introduction

It is widely accepted that bone and muscle interact mechanically as movement of the skeleton by muscle is essential for life. Less well-known but becoming more generally accepted is that muscle and bone can communicate through secreted factors (*Brotto and Bonewald, 2015*; *Bonewald, 2019*). Muscle produces factors such as β-aminoisobutyric acid and irisin with exercise, that have positive effects on bone, adipose tissue, brain, and other organs, whereas sedentary muscle produces factors such as myostatin that has negative effects on both bone and muscle (*Brotto and Bonewald, 2015*; *Karsenty and Mera, 2018*; *Kitase et al., 2018*; *Boström et al., 2012*; *Hamrick et al., 2006*).

Many of the factors secreted by bone are produced by osteocytes, the most abundant and the longest-living bone cell (*Bonewald, 2011*; *Dallas et al., 2013*). These cells are derived from terminally differentiated osteoblasts that become surrounded by the newly mineralizing bone matrix (*Dallas et al., 2013*). Osteocytes are multifunctional and appear to be the major mechanosensory cell in bone (*Bonewald, 2011*; *Temiyasathit and Jacobs, 2010*; *Uda et al., 2017*). Under unloaded conditions, these cells produce sclerostin, a negative regulator of bone formation and receptor activator of nuclear factor kappa β ligand (RANKL), the major factor that recruits and activates osteoclasts to resorb bone (*Nakashima et al., 2011*; *Xiong and O'Brien, 2012*; *Xiong et al., 2015*; *Ono et al., 2020*). In contrast, with anabolic mechanical loading, these cells produce factors such as prostaglandin E2 that have positive effects on myogenesis and muscle function (*Mo et al., 2015*). Osteocytes play a major role in mineral metabolism, through regulation of both calcium and phosphate homeostasis. Osteocytes secrete fibroblast growth factor 23 to target the kidney to regulate phosphate excretion. Both parathyroid hormone (PTH) and parathyroid-related peptide (PTHrP) regulate calcium homeostasis via the PTH type 1 receptor on osteocytes (*Feng et al., 2009*; *Teti and Zallone, 2009*). Under the physiological calcium-demanding condition of lactation, osteocytes respond to PTHrP by removing their surrounding perilacunar matrix to provide calcium for offspring, and upon weaning this perilacunar matrix is rapidly replaced, a process referred to as perilacunar remodeling (*Qing and Bonewald, 2009*; *Qing et al., 2012*; *Wysolmerski, 2013*). However, under pathological conditions such as ovariectomy, hyperparathyroidism, hypophosphatemic rickets, and cancer, excessive removal of their perilacunar matrix occurs through osteocytic osteolysis (*Tsourdi et al., 2018*; *Jähn-Rickert and Zimmermann, 2021*; *Pin et al., 2021*; *Shimonty et al., 2023*).

Bone is the largest calcium reservoir in the body and human mothers can lose an average of 250 mg/day of calcium in milk, emphasizing the need for a calcium-replete diet to prevent bone loss (*Qing et al., 2012*; *Wysolmerski, 2002*; *Kalkwarf, 2004*). During lactation, PTHrP targets the osteocyte to elevate genes coding for factors necessary for the removal of their calcium-ladened perilacunar matrix and to increase RANKL as an activator of osteoclasts (*Kovacs, 2001*). During lactation, RANKL targets osteoclasts, thereby driving osteoclastic bone resorption. Osteocytic osteolysis is accomplished through the expression of 'osteoclast-specific' genes such as cathepsin K (*Ctsk*), tartrate-resistant acid phosphatase (TRAP, gene *Acp5*), and carbonic anhydrase 1 (*Car 1*) (*Qing and Bonewald, 2009*; *Qing et al., 2012*). In addition, there is an increase in genes coding for the proton pumps, ATPase H$^+$ transporting V1 subunit G1 (*Atp6v1g1*), and ATPase H$^+$ transporting V0 subunit D2 (*Atp6v0d2*) necessary to dissolve and remove calcium from bone collagen (*Jähn et al., 2017*).

Systemic calcium deficiency such as a decrease in dietary calcium triggers an increase in PTH, acting to mobilize calcium from bones to maintain normal homeostatic circulating calcium (*Goltzman, 2008*). Worldwide, over 3.5 billion people suffer from dietary calcium deficiency, and women are at a higher risk of this condition (*Kumssa et al., 2015*; *Body et al., 2016*). Aging often results in hypocalcemia and bone loss due to low vitamin D, hypoparathyroidism, genetic abnormalities, medications decreasing dietary calcium absorption, and menopause in women. Calcium deficiency can lead to osteopenia, osteoporosis, and increased fracture risk, primarily due to secondary hyperparathyroidism (*Kumssa et al., 2015*; *Body et al., 2016*).

Irisin is a recently discovered myokine generated in response to exercise when fibronectin type III domain containing protein 5 (FNDC5) is proteolytically cleaved by a yet undetermined protease (*Boström et al., 2012*). FNDC5 is expressed in the heart, kidney, testes, brain, and other tissues; however, skeletal muscle appears to be the primary producer (*Erickson, 2013*; *Maak et al., 2021*; *Tsourdi et al., 2022*). Cleaved irisin circulates to distant organs, such as adipose tissue where irisin increases a thermogenic gene program, including the expression of uncoupling protein 1 in a process referred to as browning. This is associated with increased energy expenditure and improvement in

glucose tolerance, both of which are important for the prevention of type 2 diabetes and the reduction of complications from obesity (*Perakakis et al., 2017*; *Korta et al., 2019*). Irisin can also regulate glucose uptake in skeletal muscle (*Lee et al., 2015*), and increases myogenesis and oxidative metabolism, responsible for increasing skeletal muscle mass (*Colaianni and Grano, 2015*). Irisin also plays an important positive role in cognitive functions with exercise, aging, and degenerative diseases such as Alzheimer's disease and Parkinson's disease (*Islam et al., 2021*). Using the tail-vein injection method to deliver exogenous irisin, it was shown that irisin can cross the blood-brain barrier (*Islam et al., 2021*).

Results from studies regarding the effects of irisin on the skeleton are complex and somewhat contradictory. *Colaianni et al., 2015* have shown that recombinant irisin exerts a beneficial effect on cortical bone in young male mice by reducing the secretion of osteoblast inhibitors and increasing the activity of osteogenic cells. However, another study has shown that recombinant irisin treatment of MLO-Y4 osteocyte-like cells induces gene- and protein-level expression of *Sost*/sclerostin, a negative regulator of bone formation while maintaining cell viability under oxidative stress (*Kim et al., 2018*). *Estell et al., 2020* have shown using female FNDC5 overexpressing female mice that irisin acts directly on osteoclast progenitors to increase differentiation and promote bone resorption. *Kim et al., 2018* have shown that 9-month-old ovariectomized FNDC5 global knockout (KO) mice are protected against ovariectomy-induced trabecular bone loss through the inactivation of osteocytic osteolysis and osteoclastic bone resorption. The majority of these studies used only male or female mice, suggesting a sex-dependent response may be responsible for these seemingly opposing findings (*Estell et al., 2020*; *Colaianni et al., 2017*; *Kawao et al., 2018*; *Ma et al., 2018*; *Colucci et al., 2021*; *Posa et al., 2021*).

As shown previously, FNDC5 deletion has a protective effect against ovariectomy-induced bone loss via a reduction of osteocytic osteolysis and osteoclastic resorption (*Kim et al., 2018*). We, therefore, hypothesized that FNDC5 deletion would also be protective against bone loss due to calcium deficiency that occurs with lactation and a calcium-deficient diet. Our data show that the female skeleton in FNDC5 null female mice was resistant to bone loss due to both lactation and low calcium. However, for FNDC5 null males, deletion not only failed to protect but exacerbated bone loss in response to low calcium. We propose that male and female osteocytes respond to irisin differently under calcium-demanding conditions based on the divergence of the male and female osteocyte transcriptome with sexual maturity when the female osteocyte must serve a critical role in reproduction and lactation.

## Results

### With lactation, FNDC5 global KO mice lose less bone and are mechanically stronger compared to WT

No significant differences were observed in either bone composition or morphometry between 4- and 5-month-old virgin wildtype (WT) and FNDC5 global KO female mice (*Figure 1A, B, and C*, detailed result in *Supplementary file 1*), showing that the absence of FNDC5/irisin does not affect female bone development. It has been previously shown that during lactation, maternal bones release calcium to supplement milk, especially in response to the large calcium demand induced by large litter size or a calcium-deficient diet (*Wysolmerski, 2002*; *Ardeshirpour et al., 2015*). Similar to previous studies, 2 weeks of lactation resulted in bone loss in both WT and KO mice, with a significant reduction in cortical bone area (Ct. B.Ar), cortical bone area fraction percentage (Ct.B.Ar/T.Ar%), and cortical thickness (Ct. Th) (*Figure 1A and B*) as well as bone mineral density (BMD) (*Figure 1C*). However, the KO mice lost less bone compared to the WT mice, as evidenced by the significantly higher bone area fraction percent, cortical thickness, and BMD (*Figure 1A, B, and C*) as well as the lower percentage of bone loss (*Supplementary file 1*). These data suggest that the FNDC5 KO mice are more resistant to the effects of calcium demand. Analysis of trabecular bone parameters including trabecular bone volume fraction (BV/TV), trabecular thickness (Tb. Th), trabecular spacing (Tb. Sp), and trabecular number (Tb. N) showed no significant difference in bone loss between lactating WT and lactating KO mice (*Supplementary file 1*). There was no significant difference in the pup numbers between WT and KO females (*Figure 1—figure supplement 1A*).

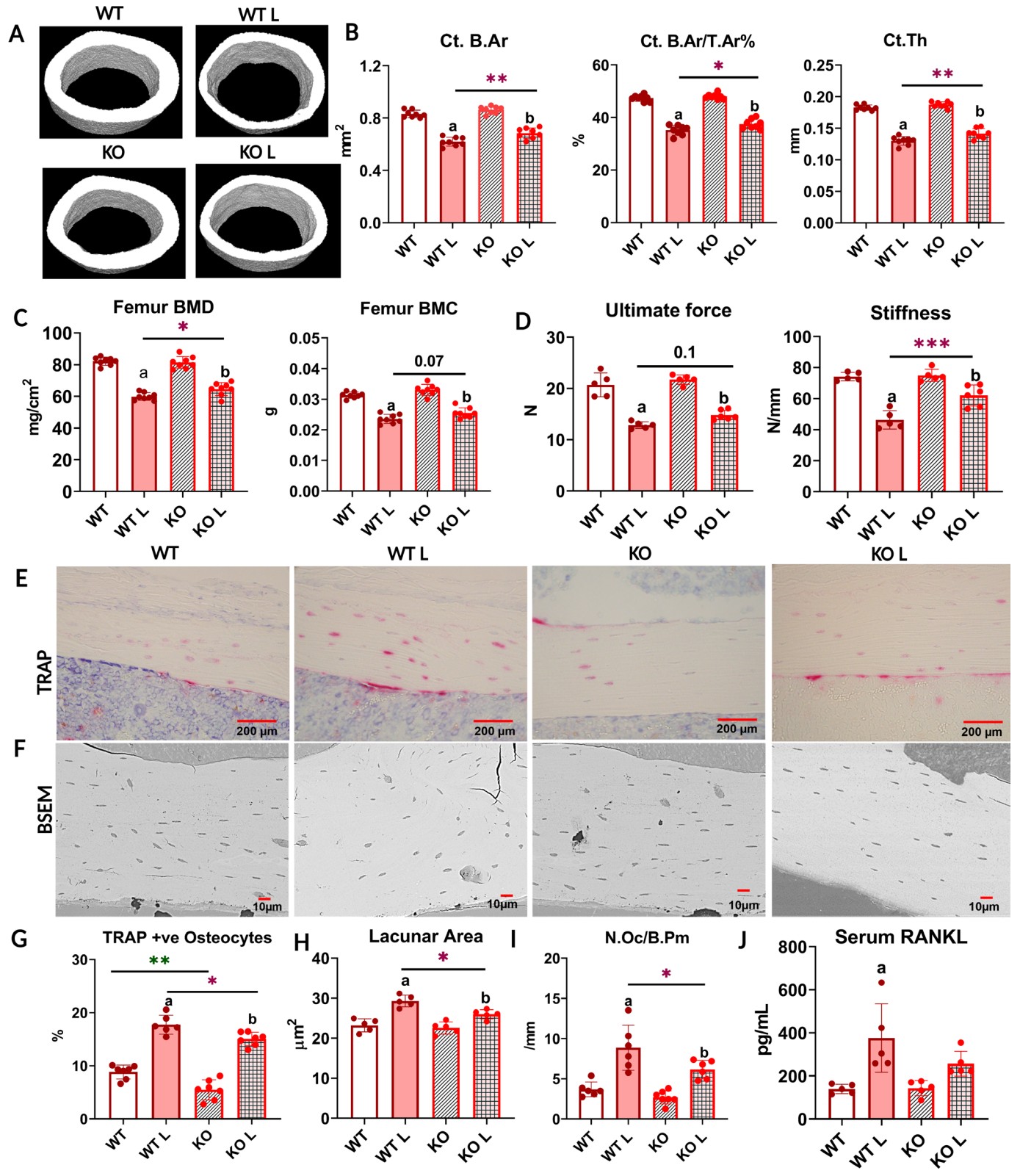

**Figure 1.** With lactation, FNDC5 global knockout (KO) mice lose less bone and are mechanically stronger compared to WT. (**A**) Respective micro-computed tomography (μCT) images of femoral midshafts from WT virgin (WT), KO virgin (KO), WT lactation (WT L), and KO lactation (KO L) mice. (**B**) μCT analysis of femoral cortical bone parameters of virgin and lactating WT and KO female mice reported as cortical bone area (Ct. B.Ar), cortical bone area fraction (Ct. B.Ar/ T.Ar %), and cortical thickness (Ct. Th). (**C**) Ex vivo dual-energy X-ray absorptiometry (DXA) analysis for bone mineral

*Figure 1 continued on next page*

*Figure 1 continued*

density (BMD) and bone mineral content (BMC) of femurs from virgin and lactating WT and KO female mice. (**D**) Three-point bending analysis of WT and KO virgin and lactating mice reported as ultimate force and stiffness. (**E**) Representative tartrate-resistant acid phosphatase (TRAP)-stained images of cortical bone from WT virgin (WT), WT lactation (WT L), KO virgin (KO), and KO lactation (KO L) mice. (**F**) Representative backscatter scanning electron microscope (BSEM) images of WT virgin (WT), KO virgin (KO), WT lactation (WT L), and KO lactation (KO L) mice femur at ×400 magnification. (**G**) Percent TRAP-positive osteocytes (TRAP+ve) in tibia from virgin and lactating WT and KO mice. (**H**) Osteocyte lacunar area in femurs from virgin and lactating WT and mice. (**I**) Osteoclast number per bone perimeter in tibia from virgin and lactating WT and KO mice. (**J**) Serum receptor activator of nuclear factor kappa β ligand (RANKL) levels in virgin and lactating WT and KO mice. 4- to 5-month-old WT and KO virgin and lactating mice, n = 5–8/group. a=Significantly different from WT, b=significantly different from KO, *=p<0.05, **=p<0.01, ***=p<0.001. Two-way analysis of variance (ANOVA) was performed for statistical analysis. The interaction was not significant.

The online version of this article includes the following figure supplement(s) for figure 1:

**Figure supplement 1.** Pup numbers for the lactation experiment, and body weight measurements for the low-calcium diet experiment.

Bone loss can have significant effects on bone mechanical properties including bone strength, stiffness, and fragility. To determine mechanical properties, three-point bending tests were performed on mice femurs. There was no significant difference between virgin WT and KO mice in terms of ultimate force and stiffness (*Figure 1D*). However, femurs from the lactating KO mice were stronger than lactating WT, as evidenced by the higher stiffness and significantly higher ultimate force needed to break the bone (*Figure 1D*, *Supplementary file 1*). This data indicates that lactating KO female bone retains greater resistance to fracture than lactating WT mice by less lactation-induced bone loss.

## With lactation, FNDC5 global KO mice have fewer TRAP-positive osteoclasts and osteocytes as well as smaller osteocyte lacunar area compared to WT mice

Previously it was shown that lactation-induced bone loss occurs via not only osteoclastic bone resorption but also osteocytic osteolysis (*Qing et al., 2012*). To determine the relative contribution of each means of resorption, tibial longitudinal sections were stained for TRAP-positive multinucleated osteoclasts as well as TRAP-positive osteocytes.

Virgin FNDC5 KO female mice had fewer TRAP-positive osteocytes compared to virgin WT mice (*Figure 1E and G*). This is the first and only difference we have observed between WT and KO female mice and suggests that the osteocytes in the female KO mice are less 'primed' to initiate osteocytic osteolysis. With lactation, TRAP-positive osteocytes significantly increased in both WT and KO mice (*Figure 1G*, detailed result in *Supplementary file 1*). Virgin KO mice started with a lower number of TRAP- positive osteocytes compared to virgin WT, and with lactation, their number of TRAP-positive osteocytes was still significantly lower compared to lactating WT (*Figure 1G*).

During lactation, in response to calcium demand, osteocytes can remove their perilacunar matrix. This process is similar but not identical to osteoclastic bone resorption (*Tsourdi et al., 2018*; *Bélanger, 1969*; *Wysolmerski, 2012*) as osteoclasts generate resorption pits, whereas osteocytes increase their lacunar size (*Qing et al., 2012*; *Wysolmerski, 2013*). We measured the osteocyte lacunar area and found no significant difference between virgin WT and KO female mice (*Figure 1F and H*) even though the KO females have fewer TRAP-positive osteocytes (*Figure 1G*). With lactation, the lacunar area increased in both groups; however, KO mice had significantly smaller average lacunar area compared to WT (*Figure 1H*). We did not observe any difference in the osteocyte density among any of the groups (WT = 258.2 ± 51.46, WT L = 274.6 ± 57.37, KO = 254.8 ± 47.66, and KO L = 273.4 ± 59.75). These data show that female lactating FNDC5 KO mice undergo less osteocytic osteolysis compared to WT females under the calcium-demanding condition of lactation.

In virgin mice, there were no significant differences in osteoclast number per bone perimeter (Oc/B.Pm) between WT and KO female mice (*Figure 1I*). With lactation, osteoclast number increased in both groups, however, KO mice had significantly fewer osteoclasts (*Figure 1I*) and a significantly lower percentage increase in the number of osteoclasts compared to WT (*Supplementary file 1*). This suggests that with lactation, fewer osteoclasts are activated in the KO as compared to the WT mice.

RANKL, another major factor in bone resorption (*Xiong and O'Brien, 2012*), is also increased during lactation to induce osteoclastic bone resorption (*Ardeshirpour et al., 2015*) by osteocytes, the major source of RANKL (*Nakashima et al., 2011*; *Xiong and O'Brien, 2012*; *Ono et al., 2020*). Virgin WT and KO mice had comparable serum RANKL levels (*Figure 1J*). With lactation, the increase

in serum RANKL was significant in the WT mice, but not in the KO mice (*Figure 1I*, *Supplementary file 1*).

## FNDC5 KO female and male bone have opposite responses to a low-calcium diet

After observing that bones are partially protected against lactation-induced bone loss in FNDC5/irisin KO female mice, we sought to determine if FNDC5/irisin null (KO) male bone is protected from calcium deficiency. Therefore, both female and male mice were placed on a calcium-deficient diet for 2 weeks to induce bone loss. We do not see any significant difference in body weight in any of the groups Figure (*Figure 1—figure supplement 1*) , or in food intake (per day average food intake was 3.9±0.9 g for WT females on a normal diet, 3.74±1.01 g for KO females on a normal diet, 3.66±1.1 g for WT females on a low-calcium diet, 3.8±0.7 g for KO females on a low-calcium diet, 4.2±1.3 g for WT males on a normal diet, 4.3±1.5 g for KO males on a normal diet, 3.94±1.8 g for WT males on a low-calcium diet, and 4.4±1.2 g for KO males on a low-calcium diet).

With regard to the female mice, similar results were observed with the low-calcium diet as was observed with lactation. At baseline, WT and KO female mice showed no significant differences in their BMD and bone mineral content (BMC) (detailed results in *Supplementary file 2*), as well as no differences in either cortical (*Figure 2B*) or trabecular bone parameters (*Supplementary file 2*). After 2 weeks of a low-calcium diet, both WT and KO female mice lost bone as can be evidenced by decreased BMD (*Supplementary file 2*) and bone area fraction (*Figure 2B*). However, similar to the lactation experiment, the KO female mice were partially resistant to bone loss compared to the female WT mice given a low-calcium diet (*Figure 2A and B*). Interestingly a higher marrow cavity area was observed in the WT compared to the KO, unlike the lactation experiment (*Supplementary file 2*). Mechanical testing showed that bone from female KO mice required a significantly higher force to break, and thus were stronger compared to WT females given a low-calcium diet (*Figure 2C*). Therefore, similar to the calcium-demanding conditions of lactation, on a low-calcium diet, the female KO bone is more resistant to bone loss than WT.

Unlike female bone, significant differences were observed between WT and KO male bone at baseline. KO male mice on a normal diet had a significantly higher BMD, BMC (*Supplementary file 2*), and bone area fraction compared to WT males of the same age (*Figure 2E*). However, femurs from KO mice had significantly lower stiffness than WT (*Figure 2F*), indicating a difference in the material properties of the bone. Therefore, the KO males have larger, denser, but weaker bones compared to WT males. To determine the effect of calcium deficiency on male mice, KO and WT mice were subjected to a low-calcium diet for 2 weeks. Unlike the female KO mice which were protected from the effects of a low-calcium diet, the KO male mice had an opposite response. The male KO mice had greater bone loss compared to the WT male mice (*Figure 2D and E*, *Supplementary file 2*), the trabecular bone loss followed the same trends but was not statistically significant (*Supplementary file 2*), and the femurs from the KO male mice were significantly less stiff and therefore weaker compared to the WT males on a low-calcium diet (*Figure 2F*). These data confirm a sex-specific response to a low-calcium diet.

To ensure that the effects observed in the KO mice were due to circulating irisin, and not FNDC5 deletion, we injected AAV8-irisin in KO male mice, with AAV8-GFP as the control, and placed them on the same low-calcium diet. We chose male mice due to the highly significant effect on bone mass and strength we saw in the KO males compared to WT males on a low-calcium diet. The irisin injection rescued the skeletal phenotype in KO male mice, shown by the higher cortical bone area fraction and the lower endosteal perimeter (*Figure 2G*). There was a tendency for higher ultimate force and stiffness in the KO males that received the AAV8-irisin injection, however, this did not reach statistical significance (*Figure 2H*). These data show that the observed effects in the FNDC5 null animals are due to an absence of irisin.

## Osteocytes from female and male KO mice respond differently to a low-calcium diet

To investigate if the bone loss was due to osteoclast or osteocyte activation, tibiae from all the groups were TRAP-stained. Under a normal control diet, the tibia from both KO female and male (*Figure 3A*)

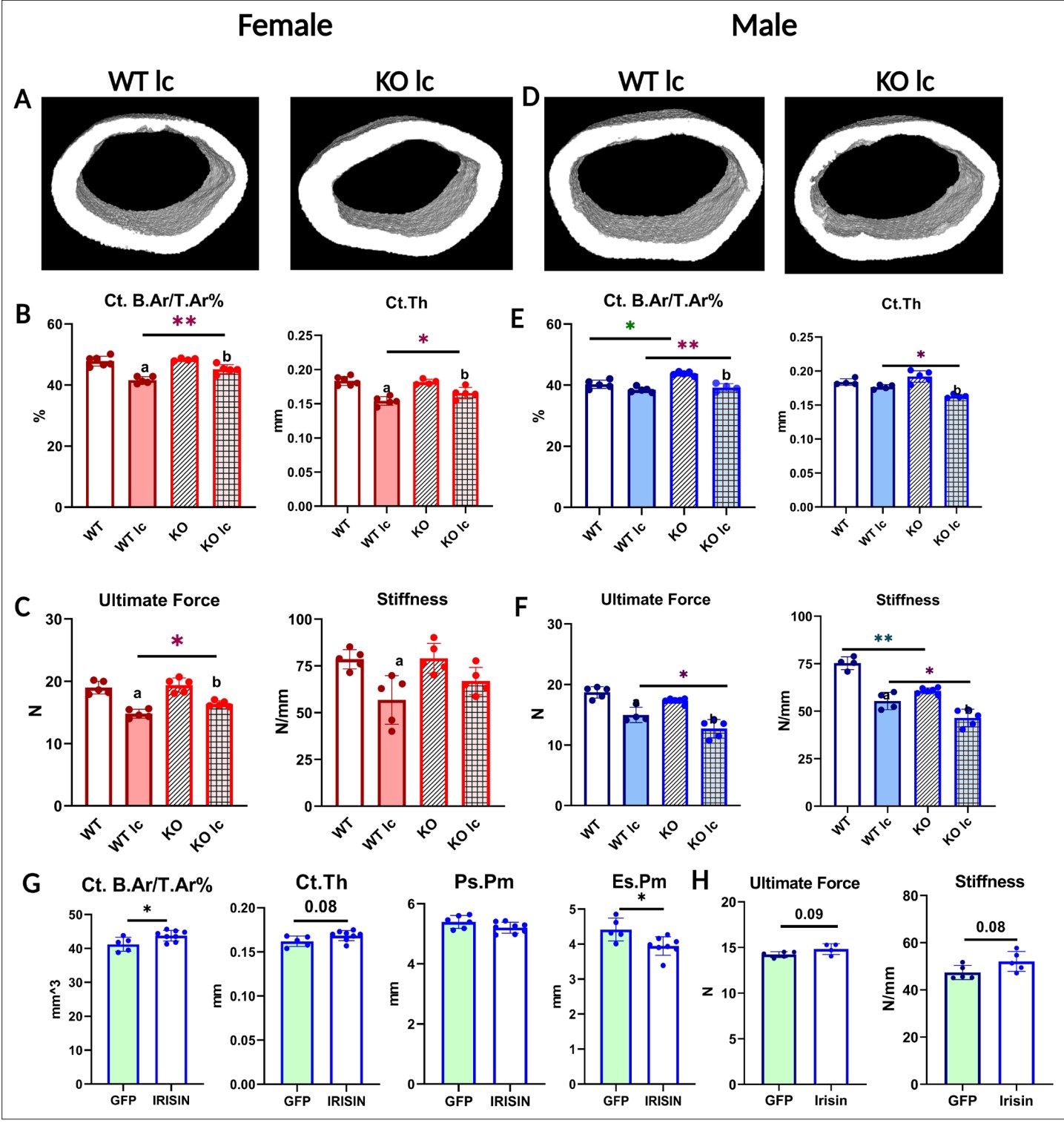

**Figure 2.** FNDC5 KO female and male mice have opposite responses to a low- calcium diet with regard to bone composition, structure, and mechanics, and irisin injection rescues FNDC5 KO male mice phenotype under a low-calcium diet. (**A**) Representative micro-computed tomography (μCT) images of femoral midshaft cortical bones from WT low-calcium diet female mouse (WT lc) and KO low-calcium diet female mouse (KO lc). (**B**) Female femoral midshaft cortical bone parameters of WT control (WT), WT low-calcium diet (WT lc), KO control (KO), and KO low-calcium diet (KO lc) mice reported as cortical bone area fraction (Ct. B.Ar/T.Ar%) and cortical thickness (Ct.Th). (**C**) Mechanical properties of femurs from female WT and KO control and low-calcium diet reported as ultimate force and stiffness. (**D**) Representative μCT images of femoral midshaft cortical bones from WT low-calcium diet male mice (WT lc) and KO low-calcium diet male mice (KO lc). (**E**) Male femoral midshaft cortical bone parameters of WT control (WT), WT low-calcium

*Figure 2 continued on next page*

Figure 2 continued

diet (WT lc), KO control (KO), and KO low-calcium diet (KO lc) mice reported as cortical bone area fraction (Ct. B.Ar/T.Ar%) and cortical thickness (Ct. Th). (**F**) Mechanical properties of femurs from male WT and KO control and low-calcium diet reported as ultimate force and stiffness. n = 4–5/group. a=Significantly different from WT, b=significantly different from KO, *=p<0.05, **=p<0.01. Two-way analysis of variance (ANOVA) was performed. As depicted here, red is female, and blue is male. (**G**) μCT measurement of femoral cortical bone of AAV8-GFP or AAV8-irisin-injected male KO mice after a 2-week low-calcium diet, reported as cortical bone area fraction (Ct. B.Ar/T.Ar%), cortical thickness (Ct. Th), periosteal parameter (Ps.Pm), and endosteal parameter (Es.Pm). (**H**) Mechanical properties of femurs from male KO low-calcium diet mice injected with AAV8-GFP or AAV8-irisin reported as ultimate force and stiffness. n = 5–7/group, *=p<0.05. Student's t-test was performed for statistical analysis between male KO GFP vs irisin-injected mice. As depicted here, green shaded bars represent GFP-injected mice.

mice had fewer TRAP-positive osteocytes compared to their WT counterparts. This indicates that their osteocytes were less 'primed' or 'activated' for resorption.

Under a low-calcium diet, the number of TRAP-positive osteocytes increased in both WT and KO female mice, similar to lactation (*Figure 3A*, *Table 1*); however, the total number was still significantly lower in the KO females than the WT females. The low-calcium diet increased TRAP-positive osteocytes in both WT and KO male mice. The KO male mice had a significantly higher level of increase (*Figure 3A*, *Table 1*), and had significantly higher TRAP-positive osteocytes compared to WT. This indicates an increased activation of osteocytes in the KO males and suggests higher osteocytic bone resorption.

There was no significant difference between WT and KO mice in osteoclast numbers per bone perimeter for both females and males (*Figure 3B*). Both WT and KO females had an increase in their multinucleated TRAP-positive osteoclast number with a low-calcium diet, however, KO females had a significantly lower number of osteoclasts compared to WT females on a low-calcium diet (*Figure 3B*). Similarly, under a normal diet, there was no difference in the number of osteoclasts between male WT and KO. Under a low-calcium diet, osteoclast numbers increased in both groups, however, there was no significant difference between WT and KO male mice (*Figure 3B*). We also measured osteoblast numbers per bone perimeter. There was no difference in osteoblast numbers in either female or male normal or low-calcium diet mice groups (data not shown).

Under normal control diet conditions, female WT mice had significantly higher osteocyte lacunar area compared to WT males (*Figure 3C and D*). There was no significant difference between FNDC5 KO female and male mice with regard to osteocyte lacunar area. This indicates that under control conditions, female osteocytes have more resorptive activity. On a low-calcium diet, all the groups have increased osteocyte lacunar area, indicating an increased level of osteocytic osteolysis (*Figure 3E*). However, in female KO mice, the average lacunar area is significantly less than in WT female mice, similar to what was observed with the lactation response. The male KO mice, on the other hand, have significantly larger lacunar areas compared to WT males on a low-calcium diet, suggesting increased osteocytic osteolysis. Together these data show that bones from female KO mice are more resistant to calcium-demanding conditions, but the deletion of FNDC5/irisin from males makes them more susceptible to bone loss under calcium-demanding conditions. This also shows that male and female KO mice respond completely differently to the challenge of calcium deficiency.

Serum RANKL levels increased in all the low-calcium diet groups compared to control diet groups (*Figure 3F*). There was no significant difference between WT and KO female mice and between WT and KO male mice. Serum PTH was measured because decreases in serum calcium stimulate the parathyroid gland to release PTH to remove calcium from bone to maintain normal calcium levels (*Jähn et al., 2017*; *Matikainen et al., 2021*). PTH levels significantly increased in WT females and WT and KO males when subjected to a low-calcium diet compared to the control diet (*Figure 3G*), however, the KO female group did not have a statistically significant increase in PTH levels. There was no significant difference in serum calcium levels in any of the groups (8–10 mg/dL range for all groups), which indicates that the elevated PTH is maintaining normal circulating calcium levels in these mice (*Figure 3H*).

Since FNDC5/irisin is robustly produced in skeletal muscle, we wanted to determine if the deletion of FNDC5/irisin affects muscle function, under either a normal or a low-calcium diet. In vivo and ex vivo muscle contractility functions were performed in these mice. No difference was found between WT and KO mice on either a normal or a low-calcium diet (*Figure 3—figure supplement 1*). This indicates deletion of FNDC5 is not affecting muscle function and that bone resorption is releasing

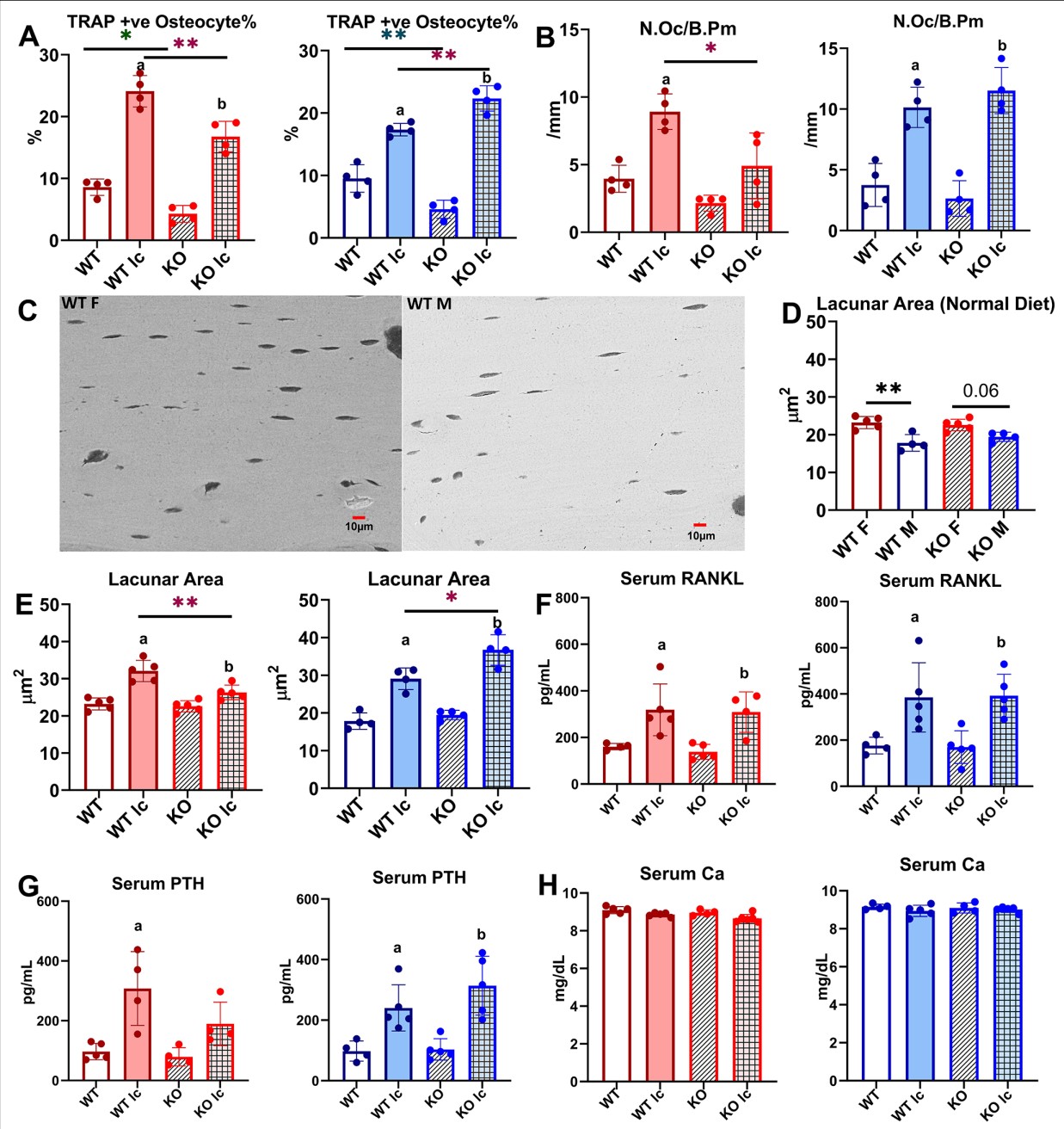

**Figure 3.** Osteocytes from female and male knockout (KO) mice respond differently to a low-calcium diet. (**A**) Percentage of tartrate-resistant acid phosphatase (TRAP)-positive (+ve) osteocytes in female and male wildtype (WT) and KO mice given a normal or a low-calcium diet. (**B**) Osteoclast number (N.Oc/B.Pm) in WT and KO female and male mice given a normal or a low-calcium diet. (**C**) Representative backscatter scanning electron microscope (BSEM) images depicting osteocyte lacunar area in femurs from WT female (WT F) and WT male (WT M) given a normal diet at ×450 magnification. (**D**) Osteocyte lacunar area in WT and KO female and male mice given a normal diet. (**E**) Lacunar area in female and male WT and KO mice given a normal or a low-calcium diet. (**F**) Serum receptor activator of nuclear factor kappa β ligand (RANKL) levels in female and male WT and KO mice given either a normal diet or a low-calcium diet. (**G**) Serum parathyroid hormone (PTH) levels in female and male WT and KO mice given either a normal diet or a low-calcium diet. (**H**) Serum calcium levels in female and male WT and KO mice given either a normal diet or a low-calcium diet. n = 4–5/group. a=Significantly different from WT, b=significantly different from KO, *=p<0.05, **=p<0.01. Two-way analysis of variance (ANOVA) was performed. As depicted here, red is female, and blue is male.

The online version of this article includes the following figure supplement(s) for figure 3:

**Figure supplement 1.** Neither genotype nor dietary calcium alters muscle functions in vivo or ex vivo.

**Table 1.** FNDC5 knockout (KO) female and male mice have opposite responses to a low-calcium diet compared to wildtype (WT) female and male mice where female KO mice are protected but male KO mice have greater bone loss than WT.

Percentage changes in different bone and serum parameters of WT and KO female and male mice with a 2-week low-calcium diet. *=p<0.05 compared to WT.

| Bone parameters and serum markers | Change | % Change in female | | % Change in male | |
|---|---|---|---|---|---|
| | | WT | KO | WT | KO |
| Bone area | Decrease | 13% | 7%* | 2% | 13%* |
| Bone area fraction | Decrease | 17% | 11%* | 7% | 23%* |
| Cortical thickness | Decrease | 19% | 13%* | 4% | 15%* |
| Osteoclast number/bone perimeter | Increase | 125% | 127% | 170% | 336%* |
| TRAP-positive osteocytes | Increase | 180% | 290% * | 85% | 388%* |
| Osteocyte lacunar area | Increase | 38% | 16% * | 60% | 89%* |
| Serum PTH | Increase | 150% | 75% * | 70% | 164%* |
| Serum RANKL | Increase | 100% | 118% | 119% | 130% |

sufficient calcium into the circulation to maintain calcium homeostasis and supplying sufficient calcium for skeletal muscle function.

## Female and male osteocyte transcriptomes are distinctly different

Total RNA sequencing of osteocyte-enriched bone chips from female and male WT mice revealed significant sex-dependent differences in the osteocyte transcriptome under normal conditions (*Figure 4A, C, and F*). The major differentially expressed genes (DEGs) were involved in the steroid, fatty acid, cholesterol, lipid transport, and metabolic processes. Compared to male WT mice, female WT mice had an approximately 2- to 3-fold higher expression of very low-density lipoprotein receptor (*Vldlr*), voltage-dependent calcium channel T type alpha 1H subunit (*Cacna1h*), aldehyde dehydrogenase (*Aldh1l2*), and a 2- to 3-fold lower expression of apolipoproteins *Apoa1*, *Apoa2*, *Apoa4*, *Apoc3* and others involved in steroid and fatty acid metabolic process. There was also a 2- to 3-fold lower expression of several lipid and solute carrier genes and apolipoprotein genes in female WT compared to male WT. This suggests that male osteocytes may be greater regulators and utilizers of these sources of energy than female osteocytes.

Differences were also observed in genes involved in extracellular matrix organization pathways, bone development, ossification, bone remodeling, and re- sorption pathways. Female WT osteocytes have higher expression of genes shown to be highly expressed in osteocytes during lactation compared to male WT osteocytes. These include *Tnfsf11* (RANKL, 2.7-fold), *Ctsk* (2.5-fold), *Acp5* (TRAP, 2.2-fold), *Mmp13* (2.7-fold), osteoclast associated receptor (*Oscar*, 4.6-fold), macrophage stimulating 1 receptor (*Mst1r*, 3-fold), as well as several collagen genes and bone formation and mineralization genes including alkaline phosphatase (*Alpl*, 2.4-fold), periostin (*Postn*, 2.6-fold), and *Dmp1* (2.2-fold). *Tgfb3* was expressed higher in the WT females compared to WT males, but no significant difference was found in either *Tgfb1* or *Tgfb2* expression levels between WT females and males. This suggests that the higher expression of bone formation genes may be to accommodate the rapid replacement of the perilacunar matrix with weaning. The upregulated and downregulated pathways in WT females compared to WT males are depicted in *Figure 4*.

## Female and male KO osteocyte transcriptomes have fewer differences compared to WT female and male transcriptomes

KO female and KO male osteocyte transcriptomes significantly differed in pathways facilitating ossification and bone mineralization, and extracellular structure and matrix organization (*Figure 4B and F*). In KO females, several collagen genes such as *Col2a1*, *Col5a2*, *Col8a2*, and *Col11a1* were 2- to 4-fold greater compared to KO males. Bone formation genes including *Alpl* (2.5-fold), osteocalcin (*Bglap*, 2.7-fold), *Postn* (2.9-fold), and *Wnt4* (2.4-fold) were also more highly expressed in KO females

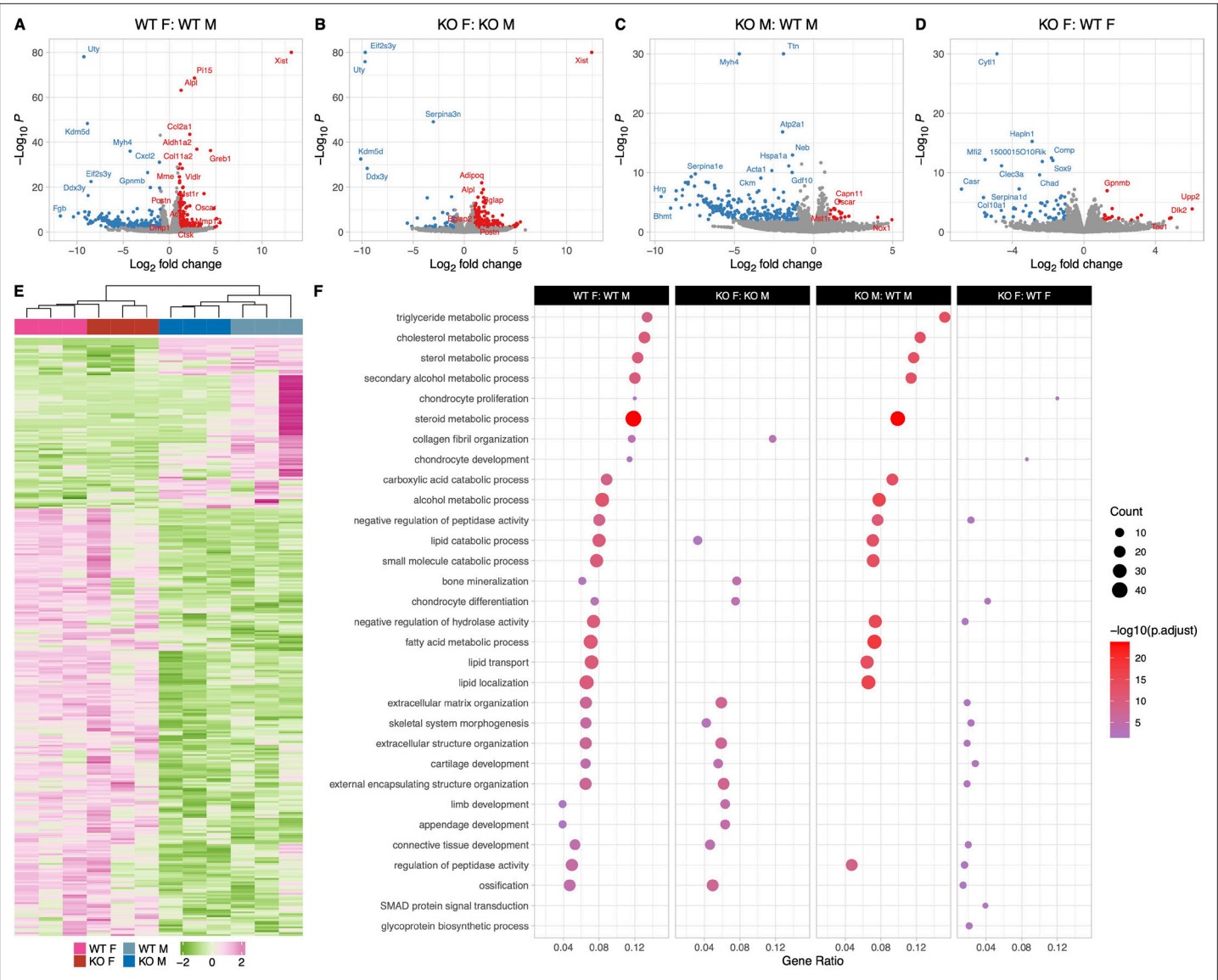

**Figure 4.** Female and male wildtype (WT) osteocyte transcriptomes are distinctly different; however, female and male knockout (KO) osteocyte transcriptomes have fewer differences compared to WT female and male transcriptomes. (**A**) Volcano plot showing the significantly regulated genes between WT female control (WT F) and WT male control (WT M) osteocyte transcriptome. (**B**) Volcano plot showing the significantly regulated genes between KO female control (KO F) and KO male control (KO M) osteocyte transcriptome. (**C**) Volcano plot showing the significantly regulated genes between WT male control (WT M) and KO male control (KO M) osteocyte transcriptome. (**D**) Volcano plot showing the significantly regulated genes between WT female control (WT F) and KO female control (KO F) osteocyte transcriptome. (**E**) Heat map showing the differentially expressed genes (DEG) among WT female control (WT F), WT male control (WT M), KO female control (KO F), and KO male control (KO M) osteocyte transcriptome. (**F**) Gene set enrichment analysis of gene ontology (GO) analysis of the significantly regulated genes between WT female control (WT F) and WT male control (WT M) osteocyte transcriptome, between KO female control (KO F) and KO male control (KO M) osteocyte transcriptome, WT male control (WT M) and KO male control (KO M) osteocyte transcriptome, and WT female control (WT F) and KO female control (KO F) osteocyte transcriptome. The figure shows the union of the top 10 GO terms of each analysis. If a term in the union, besides the top 10, is also significant (adjusted p-value less than 0.05 was used for GO analysis) in an analysis, it is also included in the figure. The latter group in the figure's title is the reference group. n = 3/group. For DEG analysis, unadjusted p-value <0.01 was used.

compared to KO males, however, the resorption genes including *Acp5* and *Ctsk* were not significantly different between KO female and KO male osteocytes. *Tgfb3* was expressed higher in the KO females compared to KO males, similar to the WTs.

The transcriptomes of WT and KO male osteocytes differed significantly, with much lower expression of genes in pathways involving steroid, fatty acid, lipid, and cholesterol transport and metabolic

processes in the KO males compared to WT males (significant genes listed in *Supplementary file 3*). A 2- to 4-fold downregulation of genes coding for solute carriers, aldehyde oxidase, and fatty acid binding proteins was observed in KO males, while *Oscar* and *Mst1r* are 2- to 3-fold higher in KO males compared to WT males. In contrast, a relatively small number of genes, 40, were differentially expressed between WT female and KO female osteocytes which reflects the lack of differences in bone morphology and bone mechanical properties (*Figure 4D and F*, *Supplementary file 3*).

## With calcium deficiency, genes responsible for osteocytic osteolysis are lower in the female KO compared to the female WT osteocyte transcriptome

Calcium deficiency in WT female mice induced higher expression of osteoclast and resorption genes compared to WT females on a normal diet (*Figure 5A and E*). *Acp5*, *Ctsk*, *Pth1r*, and *Mst1r* were elevated 2- to 4-fold in the calcium-deficient WT females. Real-time PCR analysis of osteocytes also showed an increase in *Tnsfs11*, *Acp5*, and *Ctsk gene* expression levels in the calcium-deficient WT females compared to WT females on a normal diet. There was no difference in *Sost* expression (*Figure 5—figure supplement 1D*). Additionally, five different Mmps (*Mmp13*, *Mmp15*, *Mmp2*, *Mmp16*, and *Mmp14*) were upregulated 2- to 3.5-fold in the WT calcium-deficient females. These are genes thought to play a role in osteocytic osteolysis. Bone formation and remodeling genes including *Bglap*, *Bglap2*, *Alpl*, *Wnt5a*, and *Wnt2b* were upregulated 2- to 5-fold in the WT low-calcium diet group compared to WT female normal diet group as well. These genes may be increased to provide quick bone formation upon return to normal calcium demand.

Calcium deficiency in KO female mice also induced increased expression of a number of osteoclast and resorption genes including *Ctsk* (2.8-fold), *Mmp13* (3- fold), and *Oscar* (2.6-fold) in comparison to KO female osteocytes on a normal diet (*Figure 5B and E*). However, unlike the WT osteocytes, expression levels of *Acp5* and *Pth1r* were not different in osteocytes from KO female mice on a normal diet or a low-calcium diet. Real-time PCR analysis also showed an increase in *Ctsk* gene expression level in the calcium-deficient KO females compared to KO females on a normal diet, with no significant difference in the expression levels of *Tnsfs11*, *Acp5*, and *Sost* genes (*Figure 5—figure supplement 1D*).

Next, we compared KO female mice on a low-calcium diet to WT female mice on a low-calcium diet (*Figure 5C and E*, significantly DEGs listed in *Supplementary file 4*). Several bone resorption genes were lower by 2-fold in KO females, including *Tnsfs11 and Mmp15*. Real-time PCR analysis also showed a significantly lower expression of the *Tnsfs11* gene in the calcium-deficient KO females compared to calcium-deficient WT females (*Figure 5—figure supplement 1D*). Additionally, bone formation genes including *Alpl*, *Bglap*, *Wnt2b*, *Col1a1*, *Col1a2*, and *Postn* were also approximately 2-fold lower in the KO low-calcium females compared to WT low-calcium females. This suggests that female KO osteocytes are less responsive to calcium deficiency than female WT osteocytes.

## With calcium deficiency, genes responsible for bone resorption, bone formation, and lipid metabolism are differentially regulated in the osteocyte transcriptome in male KO mice compared to male WT mice

Calcium deficiency in WT male mice caused a 2- to 7-fold increased expression of *Tnsfs11*, *Acp5*, *Ctsk*, *Oscar*, and *Mst1r* in their osteocyte transcriptome compared to WT males on a normal diet (*Figure 6A and E*). Real-time PCR validation also showed a similar increase in *Tnsfs11*, *Acp5*, and *Ctsk* gene expression levels in the calcium-deficient WT males compared to WT males on a normal diet (*Figure 5—figure supplement 1E*). Bone formation and remodeling genes including *Postn*, *Col1a1*, *Col1a2*, *Bglap*, and *Wnt4* were also elevated 2- to 4-fold in the WT male low-calcium diet compared to the WT normal diet control group.

Multiple genes involved in the steroid and fatty acid metabolic process pathways as well as lipid catabolic processes were downregulated 2- to 7-fold in the calcium-deficient WT males compared to WT males on a normal diet. These genes include several solute carrier family protein genes *Slc27a2* and *Slc27a5*, several apolipoprotein genes including *Apoa1*, *Apob*, and *Apoc1*, several cyp genes including *Cyp2e1*, *Cyp7a1*, and *Plin1*.

Similarly, osteocytes from KO males on a low-calcium diet had a 2- to 4-fold higher expression of osteoclast genes such as *Tnsfs11*, *Oscar*, and *Car3* and a 2- to 5-fold upregulation of bone formation genes such as *Col1a1*, *Col1a2*, *Alpl*, *Bglap*, and *Postn* compared to osteocytes from KO males on a

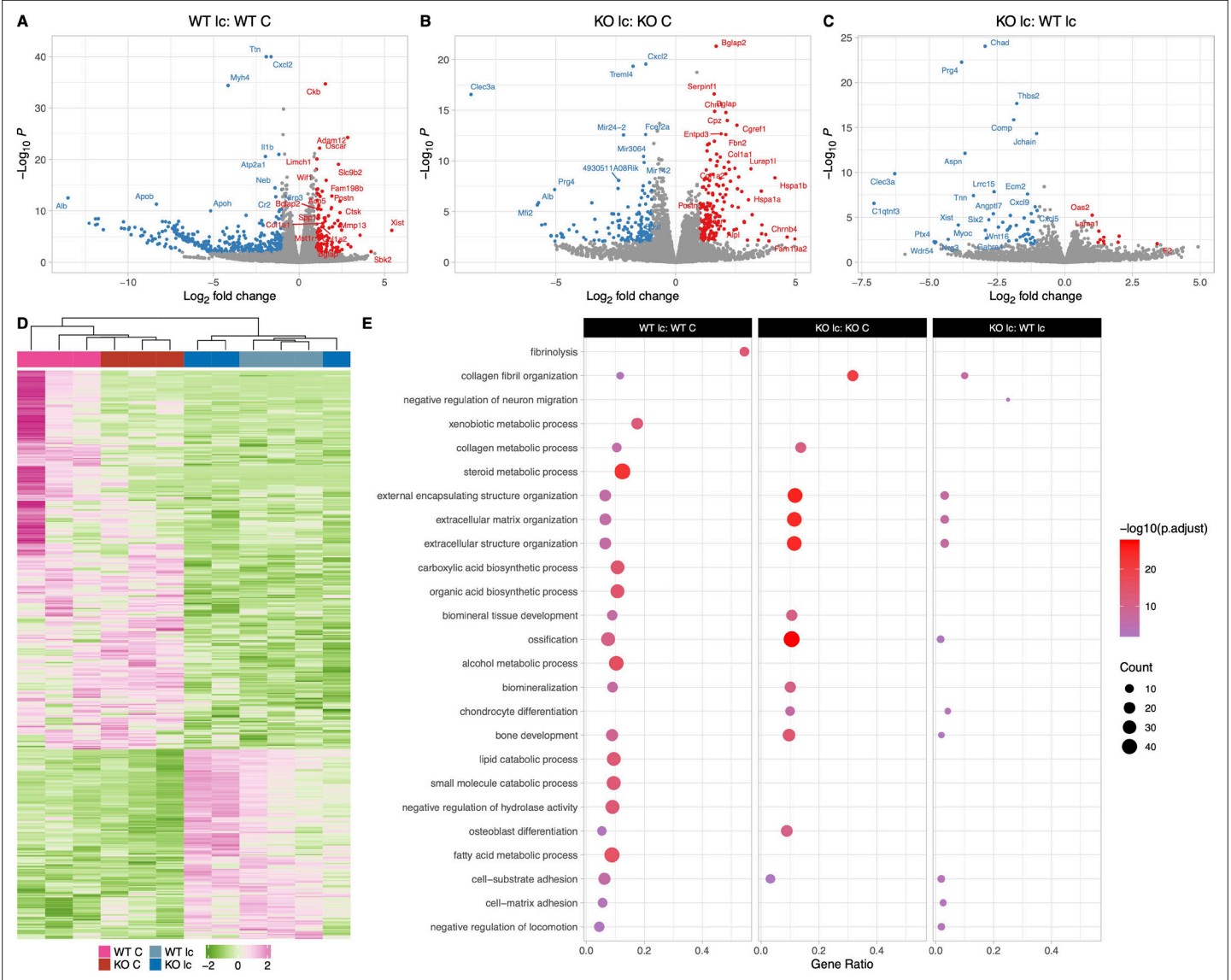

**Figure 5.** The osteocyte transcriptomes from female wildtype (WT) and knockout (KO) mice are distinct when challenged with a low-calcium diet. (**A**) Volcano plot showing the significantly regulated genes between WT female control (WT C) and WT female low-calcium diet-fed mice (WT lc) osteocyte transcriptome. (**B**) Volcano plot showing the significantly regulated genes between KO female control (KO C) and KO female low-calcium diet-fed mice (KO lc) osteocyte transcriptome. (**C**) Volcano plot showing the significantly regulated genes between WT female low-calcium diet-fed mice (WT lc) and KO female low-calcium diet-fed mice (KO lc) osteocyte transcriptome. (**D**) Heat map showing the differentially expressed genes (DEGs) among WT female control (WT C), WT female low-calcium diet-fed mice (WT lc), KO female control (KO C), and KO female low-calcium diet-fed mice (KO lc) osteocyte transcriptome. (**E**) Gene set enrichment analysis of gene ontology (GO) analysis of the significantly regulated genes between WT female control (WT C) and WT female low-calcium diet-fed mice (WT lc) osteocyte transcriptome, between KO female control (KO C) and KO female low-calcium diet-fed mice (KO lc) osteocyte transcriptome, and WT female low-calcium diet-fed mice (WT lc) and KO female low-calcium diet-fed mice (KO lc) osteocyte transcriptome. The figure shows the union of the top 10 GO terms of each analysis. If a term in the union, besides the top 10, is also significant (adjusted p-value less than 0.05 was used for GO analysis) in an analysis, it is also included in the figure. The latter group in the figure's title is the reference group. n = 2–3/group. For DEG analysis, unadjusted p-value <0.01 was used.

The online version of this article includes the following figure supplement(s) for figure 5:

**Figure supplement 1.** Quality control and validation of RNA sequencing.

normal diet (**Figure 6B and E**). Therefore, genes responsible for bone resorption and bone formation were increased in both WT and KO with calcium deficiency. Real-time PCR data showed an increase in *Tnsfs11*, *Acp5*, and *Ctsk* gene expression levels in the calcium-deficient KO males compared to KO males on a normal diet, validating the RNA sequencing data (**Figure 5—figure supplement 1E**).

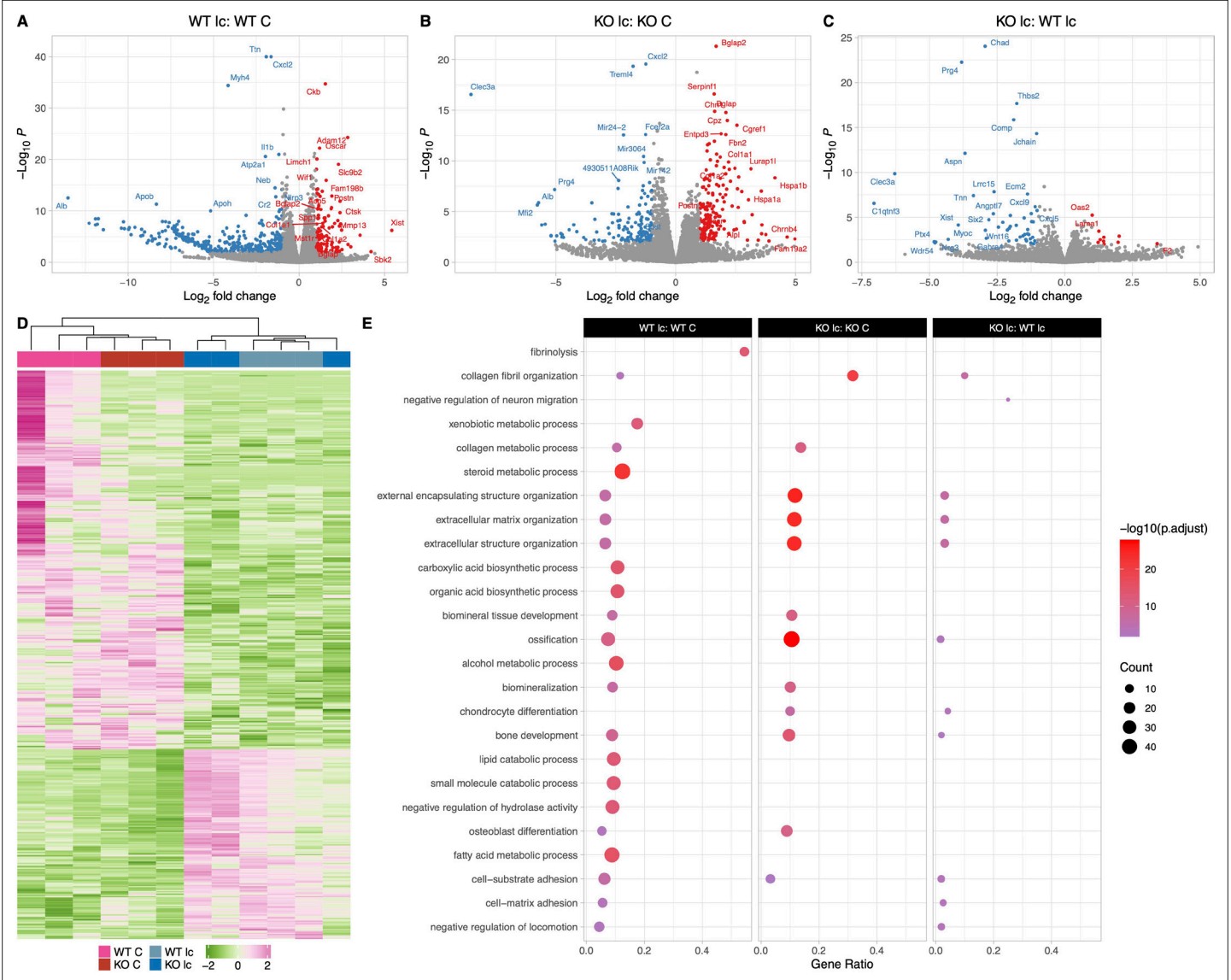

**Figure 6.** The osteocyte transcriptomes from male wildtype (WT) and knockout (KO) mice are distinct when challenged with a low-calcium diet. (**A**) Volcano plot showing the significantly regulated genes between WT male control (WT C) and WT male low-calcium diet-fed mice (WT lc) osteocyte transcriptome. (**B**) Volcano plot showing the significantly regulated genes between KO male control (KO C) and KO male low-calcium diet-fed mice (KO lc) osteocyte transcriptome. (**C**) Volcano plot showing the significantly regulated genes between WT male low-calcium diet-fed mice (WT lc) and KO male low-calcium diet-fed mice (KO lc) osteocyte transcriptome. (**D**) Heat map showing the differentially expressed genes (DEGs) among WT male control (WT C), WT male low-calcium diet-fed mice (WT lc), KO female control (KO C), and KO male low-calcium diet-fed mice (KO lc) osteocyte transcriptome. (**E**) Gene set enrichment analysis of gene ontology (GO) analysis of the significantly regulated genes between WT male control (WT C) and WT male low-calcium diet-fed mice (WT lc) osteocyte transcriptome, between KO male control (KO C) and KO male low-calcium diet-fed mice (KO lc) osteocyte transcriptome, and WT male low-calcium diet-fed mice (WT lc) and KO male low-calcium diet-fed mice (KO lc) osteocyte transcriptome. The figure shows the union of the top 10 GO terms of each analysis. If a term in the union, besides the top 10, is also significant (adjusted p-value less than 0.05 was used for GO analysis) in an analysis, it is also included in the figure. The latter group in the figure's title is the reference group. n = 3/ group. For DEG analysis, unadjusted p-value <0.01 was used.

When KO males were compared to WT males on a low-calcium diet (***Figure 6C and E***, ***Supplementary file 4***), there was a 2- to 3-fold higher expression of bone resorption genes including *Oscar* and *Mst1r* in the KO low-calcium diet males compared to WTs. Several collagen formation genes and ossification genes including *Col3a1*, *Col8a2*, *Tnn*, *Aspn*, and *Igfbp6* were also significantly downregulated in the KO males on a low-calcium diet compared to WTs on a low-calcium diet. It is not clear whether these also play a role in the increased bone resorption observed with calcium deficiency in KO males.

Real-time PCR analysis showed no significant difference in expression levels of *Tnsfs11*, *Acp5*, *Sost*, and *Ctsk* genes between calcium-deficient KO males and calcium-deficient WT males, reflecting the RNA sequencing data (*Figure 5—figure supplement 1E*). No significant difference was observed in expression levels of genes involved in the lipid catabolic process pathway or fatty acid metabolism pathways.

## Male and female osteocytes respond differently to calcium deficiency in a genotype-specific manner

In response to 2 weeks of calcium deficiency, WT female mice had higher expression of genes involved in extracellular matrix and structure organization as well as ossification compared to WT male mice with calcium deficiency. Calcium deficiency in WT female mice caused significantly increased expression of bone formation genes compared to WT males including several collagen genes such as *Col2a1*, *Col6a3*, *Col4a2*, as well as *Postn*, and *Bglap2*. This was accompanied by an increased expression of bone resorbing genes in WT females including several *Car* genes, *Mmp13*, *Mmp16*, *Tnsfs11*, and *Mst1r* in their osteocyte transcriptome compared to WT males on a low-calcium diet (*Figure 7A, C, and D*). This suggests that both bone formation and bone resorption are upregulated in WT females compared to WT males in response to calcium deficiency, and WT females undergo higher bone remodeling compared to WT males.

On the other hand, in response to calcium deficiency, KO female and male mice have less significantly differently expressed genes compared to WT females and males (*Figure 7B, C, and D*). The major upregulated bone formation genes in KO females compared to KO males include several collagen genes such as *Col2a1* and *Col8a2*. The major bone resorption genes that were upregulated in KO females compared to KO males were *Mmp13* and *Dcstamp*.

## Discussion

Irisin has been shown to be increased in the blood of humans and mice with exercise. Irisin, working mainly through its receptor $\alpha V \beta 5$ integrin, has been shown to have powerful effects on fat, bone, and brain tissues (*Boström et al., 2012*; *Tsourdi et al., 2022*; *Korta et al., 2019*; *Islam et al., 2021*; *Kim et al., 2018*; *Colaianni et al., 2017*; *Wrann et al., 2013*; *Xin et al., 2016*; *Wang et al., 2017*; *Bao et al., 2022*; *Zhang et al., 2022*). With regard to bone, studies have generated complex and even contradictory results (*Erickson, 2013*; *Maak et al., 2021*; *Colaianni and Grano, 2015*; *Kim et al., 2018*; *Estell et al., 2020*; *Colaianni et al., 2014*; *Zhang et al., 2018*). Of note, the majority of bone studies have been performed either exclusively on males, or females, but few on both. Most studies have used recombinant irisin treatment whereas we have focused on the effects of deleting irisin. Other studies have mainly examined the effects on osteoblasts and osteoclasts, whereas our studies have focused on osteocytes (*Kim et al., 2018*).

Global deletion of FNDC5 on a normal diet had essentially no effect on bone in females, but in contrast, the null male mice have significantly more bone compared to WT males, but this bone has impaired mechanical properties. This suggests that the lack of FNDC5 is having no effect on the development or growth of the female skeleton, but does affect the male skeleton, increasing the size yet impairing matrix properties responsible for strength. Examination of their osteocytes showed that both female and male null mice have significantly fewer TRAP-positive osteocytes compared to their sex-matched WT controls suggesting that their osteocytes are more quiescent or less primed for bone resorption.

Challenging the null animals with calcium deficiency revealed dramatic differences in osteocytic osteolysis and osteoclast activation, two major functions of osteocytes. Deletion of FNDC5 in females is partially protective against calcium deficiency, but deletion in males accelerates both of these osteocyte functions resulting in greater bone loss compared to controls. We have shown previously that under calcium-demanding conditions such as lactation, osteocytes express genes previously thought only to be specific for osteoclasts including cathepsin K, TRAP, carbonic anhydrase, the proton pump V-ATPase, and others (*Qing et al., 2012*) and shown that osteocytes are the major source of RANKL (*Nakashima et al., 2011*; *Xiong and O'Brien, 2012*; *Xiong et al., 2015*). In this study, lactating females lacking FNDC5 were partially resistant to bone loss, similar to ovariectomized females as previously published (*Kim et al., 2018*). To determine the effects of calcium deficiency on males, mice

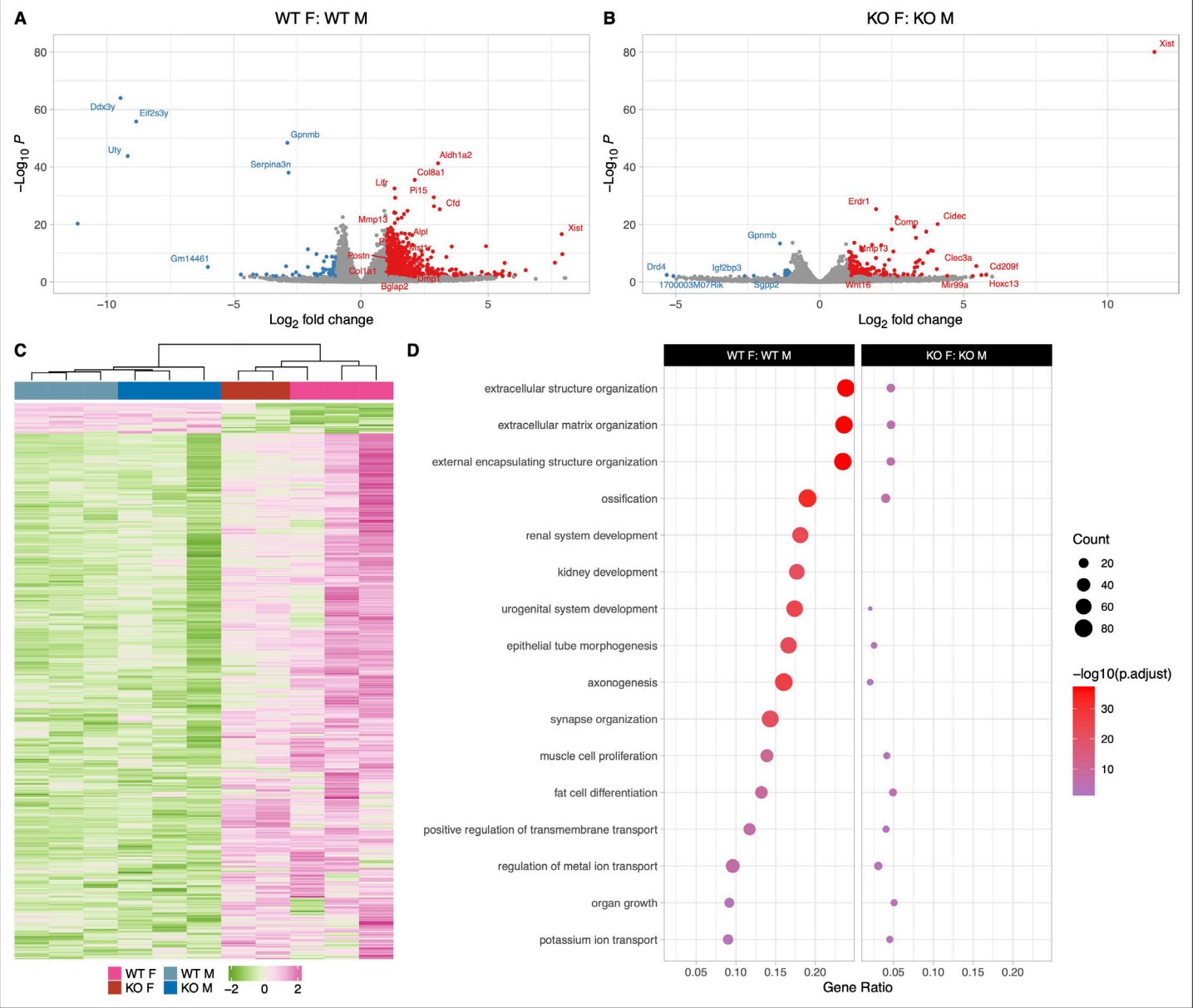

**Figure 7.** The osteocyte transcriptomes from male and female mice are distinct when challenged with a low-calcium diet. (**A**) Volcano plot showing the significantly regulated genes between WT female low-calcium diet-fed (WT F) and WT male low-calcium diet-fed mice (WT M) osteocyte transcriptome. (**B**) Volcano plot showing the significantly regulated genes between KO female low-calcium diet-fed (KO F) and KO male low-calcium diet-fed mice (KO M) osteocyte transcriptome. (**C**) Heat map showing the differentially expressed genes (DEGs) among WT male low-calcium diet-fed mice (WT M), KO male low-calcium diet-fed mice (KO M), WT female low-calcium diet-fed (WT F), and KO female low-calcium diet-fed (KO F) osteocyte transcriptome. (D) Gene set enrichment analysis of gene ontology (GO) analysis of the significantly regulated genes between WT female low-calcium diet-fed (WT F) and WT male low-calcium diet-fed mice (WT M) osteocyte transcriptome, and between KO female low-calcium diet-fed (KO F) and KO male low-calcium diet-fed mice (KO M) osteocyte transcriptome. The figure shows the union of the top 10 GO terms of each analysis. If a term in the union, besides the top 10, is also significant (adjusted p-value less than 0.05 was used for GO analysis) in an analysis, it is also included in the figure. The latter group in the figure's title is the reference group. n = 2–3/group. For DEG analysis, unadjusted p-value <0.01 was used.

were given a low-calcium diet for 2 weeks. Unlike the protective effects of FNDC5/irisin deletion in females, bone loss was exacerbated in null males compared to controls on a low-calcium diet.

With 2 weeks of lactation and litter size comparable to WT controls, the null female mice had less circulating RANKL, fewer TRAP-positive osteoclasts, fewer TRAP-positive osteocytes, and smaller lacunar size. Our observation that the deletion of FNDC5/irisin makes lactating mice partially resistant to bone loss has an important implication with regard to the purpose of lactation. Lactation is a

critical period for pups as they obtain essential nutrients, especially calcium, from the mother's milk for their proper growth. Calcium lost by the mother's bone during lactation is rapidly replaced upon weaning with complete recovery of bone mass within a week (*Qing et al., 2012*; *Wysolmerski, 2002*; *Kalkwarf, 2004*; *Kovacs, 2001*; *Wysolmerski, 2012*). Our data suggest that FNDC5/irisin acts as a regulator of calcium release from maternal bones to fulfill the offspring demands during lactation. Therefore, irisin appears to play a beneficial role in ensuring offspring survival and consequently, successful reproduction.

To determine if low calcium would have a similar effect on male FNDC5 null bone, both males and females were subjected to a low-calcium diet for 2 weeks. The effects of a low-calcium diet on female osteocytes and bone loss were essentially identical to the effects of lactation, with two exceptions. First, serum RANKL levels were not significantly different between virgin and lactating null females, while they were between null females on a normal compared to a calcium diet suggesting that RANKL plays less of a role in lactation compared to calcium deficiency. Second, the medullary cavity and endosteal bone in the low-calcium females were completely protected in the FNDC5 null females but were not in the lactating FNDC5 null mice. Bone loss due to lactation or due to dietary calcium deficiency may target different bone sites. Our unpublished observations suggest that endosteal bone is removed faster than periosteal bone with lactation, but this remains to be carefully validated. This difference may also be due to elevated PTHrP during lactation (*Kovacs, 2001*), whereas hypocalcemia increases circulating PTH levels (*Goltzman, 2008*), and it is not clear if hormones target distinct bone sites. Similar to the lactating FNDC5 null mice, the null females placed on the low-calcium diet had fewer TRAP-positive osteoclasts, fewer TRAP-positive osteocytes, and smaller lacunar size. Serum RANKL levels increased in both WT and null mice with dietary calcium deficiency, therefore, serum RANKL alone is not enough to explain the partial protective effect of FNDC5 deletion against bone loss. In summary, female null mice are not only resistant to bone loss due to estrogen deficiency as we showed previously (*Kim et al., 2018*) but are also resistant to calcium deficiency either due to an increase in PTHrP as with lactation, or an increase in PTH as with a low-calcium diet.

Osteoporosis manifests earlier in females due to menopause, but males also develop osteoporosis but at an older age (*Johannesdottir et al., 2013*; *Johnston and Dagar, 2020*), and the elderly are known to suffer from calcium deficiency which accelerates bone loss (*Kumssa et al., 2015*; *Body et al., 2016*). Dietary calcium deficiency has been shown previously to affect female and male bone differently where female rat bones are more sensitive to a low-calcium diet compared to males (*Geng and Wright, 2001*). Similarly, in our study, we saw that WT females were more affected by calcium deficiency and lost more bone compared to WT male mice. However, the opposite was observed for the FNDC5/irisin null mice, where female null mice were partially resistant, and male null mice were more susceptible to bone loss with calcium deficiency compared to their WT counterparts. Despite starting with more bone volume compared to WT, the FNDC5 null males had increased osteocyte lacunar area and lost more bone with dietary calcium deficiency compared to WT males. This greater bone loss can be explained through the dramatic increase of TRAP-positive osteocytes and TRAP-positive osteoclasts, but not by a significantly greater increase in circulating RANKL. This sex difference indicates that FNDC5/irisin may be involved in the regulation of calcium release from bone via osteocytes in a sex-dependent manner.

Lacunar area is an indicator of osteocyte regulation of their lacunar microenvironment. Here, we report that osteocyte lacunar size is significantly larger in virgin WT female mice compared to same-age WT males. This difference in lacunar area indicates a distinction between female and male osteocyte function. The mammalian skeleton is a sexually dimorphic organ (*Sharma et al., 2023*), and female and male bones respond differently to circulating factors, hormones, and myokines as well as other challenges (*Kurapaty and Hsu, 2022*; *Lu et al., 2022*; *Osipov et al., 2022*). As osteocytes are regulators of bone formation and resorption (*Bonewald, 2011*; *Dallas et al., 2013*; *Robling and Bonewald, 2020*), this sex difference may be due to differences in male and female osteocytes. A recent study by Youlten and colleagues has shown that male and female osteocyte transcriptomes are distinctly different (*Youlten et al., 2021*). At 4 weeks of age, the female osteocyte transcriptome diverges from the male osteocyte transcriptome and these differences continue with age. A cluster of genes more highly expressed in female osteocytes compared to male osteocytes are those involved in bone resorption, the same ones elevated in osteocytes in response to lactation. These transcripts include genes necessary for osteocytic perilacunar remodeling and reduction in pH, which are essential for

calcium removal (*Qing et al., 2012*). This suggests that the larger lacunar area in female osteocytes compared to male osteocytes may be due to the higher expression of bone resorption genes.

The magnitude of the effect size due to FNDC5 deficiency appears modest with regard to the quantitative cortical bone parameters. However, if one examines the changes in osteocyte lacunar size and the mechanical properties of these bones, the differences are greater. As shown in *Figure 3E*, the lacunar area of the WT females on a low-calcium diet increases by over 30% and the FNDC5 null by less than 20%, while in the males it is approximately 38% in WT compared to 46% in null. According to *Buenzli and Sims, 2015*, a potential total loss of ~16,000 mm$^3$ (16 mL) of bone occurs through lactation in the human skeleton. This was based on our measurements in lactation-induced murine osteocytic osteolysis (*Qing et al., 2012*). They used our 2D section of tibiae from lactating mice showing an increase in lacunar size from 38 to 46 μm$^2$. In that paper we also showed that canalicular width is increased with lactation. Therefore, this suggests dramatically lower intracortical porosity due to the osteocyte lacunocanalicular system in female null mice compared to female WT mice either with lactation or a low-calcium diet and a dramatic increase in intracortical porosity in null males compared to WT males on a low-calcium diet. Based on these data, using the FNDC5 null animals, we would speculate that the product of FNDC5, irisin, is having a significant effect on the ultrastructure of bone in both males and females challenged with a low-calcium diet.

To begin to understand the molecular mechanisms responsible for the sex and genotype differences, we compared the osteocyte transcriptomes of 5-month-old female and male, WT and null mice. Our results show that the osteocyte transcriptomes of female and male WT mice are significantly different under normal conditions. A surprising difference we observed but not described in the Youlten paper (*Youlten et al., 2021*) was that compared to WT female osteocytes, WT male osteocytes have much higher expression of genes involved in steroid, lipid, and cholesterol metabolism and transport pathways, lipid and solute carrier genes, and apolipoprotein genes. This suggests that osteocyte metabolism and bioenergetics are distinctly different between WT females and WT males. We hypothesize that the DEGs in these bioenergetic and metabolic pathways modulate bone mass and formation and may shed light on the sexual dimorphism of bones. As these differentially regulated pathways were not previously reported by *Youlten et al., 2021*, this may be due to differences in strain, housing, diet, or microbiome. Another explanation is the greater osteocyte purity in our study as we used a series of collagenase digestions and EDTA chelation to remove any surface cells which was not performed in the Youlten paper (*Youlten et al., 2021*).

A second major difference between female and male WT osteocytes was the higher expression of genes involved in collagen matrix formation, bone mineralization, remodeling, resorption, and osteocytic osteolysis pathways in females compared to males. Many of the highly expressed bone resorption genes in WT female osteocytes have been shown to be elevated during lactation (*Qing et al., 2012*) including *Acp5*, *Ctsk*, and *Mmp13*, all involved in osteocytic osteolysis. This further supports our hypothesis that WT female osteocytes are more primed for resorption compared to WT males, presumably to meet the increased calcium demand during lactation, and correlates with the observed larger lacunae compared to males.

TGFβ is another potential player in osteocyte perilacunar/canalicular remodeling. Alliston and colleagues generated transgenic mice with reduced expression of the TGFβ type II receptor in mice expressing Dmp1-Cre (*Dole et al., 2020*) (PMID: 32282961) and found a significant difference in bone parameters and markers of osteocyte perilacunar remodeling between the sexes. The females were subjected to lactation and the transgenics were found to be resistant to osteocytic osteolysis compared to controls. However, these investigators did not investigate the lacunar remodeling process in males as compared to females as was performed in the present study using a low-calcium diet. Their study does suggest that TGFβ is involved in the osteocytic osteolysis that occurs with lactation, however, even though the transgenic males showed a disrupted lacunocanalicular network compared to WT males, this does not necessarily indicate a defect in perilacunar remodeling. It is more likely that the defect occurred during bone formation when osteoblasts were differentiating into osteocytes. In our study, we observed a higher expression of *TGFb3* in WT female mice compared to WT male mice, with no significant differences in *TGFb1* or *TGFb2* expression. This suggests that TGFβ3 may play a role in generating the larger lacunar area in WT females compared to WT males through increased matrix-related signaling in irisin-replete conditions.

Few differences were observed between WT female and null female osteocyte transcriptomes as would be expected for bone morphometry and the only difference observed was the number of TRAP-positive osteocytes. In contrast, osteocytes from WT males and null males are significantly different with regard to fatty acid and lipid metabolism pathways whereas null male mice have lower expression of these genes compared to WT males. This suggests a role for irisin in lipid metabolism and bioenergetics in male osteocytes. Lower expression in the null male mice may be responsible for the higher bone mass and inferior biomechanical properties compared to WT males suggesting these pathways mediate the effects of FNDC5/irisin on male bone.

Osteocytes from null females have higher expression of genes and pathways involved in collagen matrix organization, ossification, and mineralization compared to null males. Unlike WT males and females, there was no difference in expression of lipid, cholesterol, and fatty acid metabolism genes in null males compared to null females. Again, this indicates that FNDC5/irisin regulates male bone through these lipid-related pathways.

Lactation and calcium deficiency induce the same changes in females. Similar to that reported previously for lactation (*Qing et al., 2012*), osteocytes from WT female mice on a low-calcium diet exhibited an increase of several osteoclast/resorption/lactation genes including *Acp5*, *Ctsk*, *Oscar*, *Mst1r*, and *Pth1r* compared to WT females on a normal diet. Surprisingly, we also observed an increase in bone formation genes including *Col1a1*, *Alpl*, and *Bglap*. As osteocytic osteolysis is rapidly reversed within a week of weaning, the osteocyte may be preparing to rapidly reverse bone loss. We propose that once calcium is replenished, shutting off the proton pump will rapidly reverse the pH within the osteocyte lacunae, allowing bone-forming proteins such as alkaline phosphatase to become active to rapidly replace the osteocyte perilacunar matrix (*Jähn et al., 2017*; *Andersson et al., 2003*; *Silver et al., 1988*; *Kaplan, 1972*; *Farley and Baylink, 1986*).

The main molecular mechanism responsible for the resistance of null female mice to calcium deficiency compared to WT female mice is lower expression of genes such as *Tnfsf11*, responsible for osteoclastic resorption. A correspondingly lower expression of bone formation genes including *Col1a1*, *Alpl*, and *Bglap* compared to WT females on a low-calcium diet was observed. The lower expression of both formation and resorption genes suggests a coupling of resorption with formation. Irisin appears to regulate calcium release in the female skeleton.

Osteocytes from WT male mice on a low-calcium diet expressed higher levels of bone resorption genes including *Tnsfs11*, *Acp5*, *Ctsk*, *Oscar*, and *Mst1r* compared to WT male mice on a normal diet as expected. Like the females, there is a coupling with bone formation genes as there is also an increase in *Bglap* and *Col1a1*, suggesting the potential for osteocytes to rapidly replace their perilacunar matrix with calcium repletion. Similarly, the male null mice with calcium deficiency showed an increase in bone resorption genes including *Tnsfs11*, *Oscar*, and *Car3*, as well as an increase in bone formation genes such as *Alpl* and *Bglap* compared to null mice on a normal diet. The major differences between WT male mice with calcium deficiency and FNDC5 null male mice with calcium deficiency were the lower expression of genes involved in the extracellular matrix organization, ossification, and bone development pathways in the null male mice compared to WT males. This suggests a mechanism for how null male mice lose more bone with calcium deficiency compared to WT males.

Irisin could be having direct or indirect effects on osteocytes. Irisin can modulate adipose tissue (*Boström et al., 2012*; *Zhang et al., 2014*; *Celi and Brown, 2017*; *Luo et al., 2022*), can potentially modulate osteogenic differentiation of bone marrow mesenchymal stem cells through $\alpha V\beta 5$ (*Zhu et al., 2023*), and bone marrow adipose tissue can modulate bone properties (*Yeung et al., 2005*; *Rosen and Bouxsein, 2006*; *Muruganandan and Sinal, 2014*; *Styner et al., 2015*; *Schwartz, 2015*; *During, 2020*) as well as osteocyte number and activity (*Al Saedi et al., 2019*; *Al Saedi et al., 2020*). Irisin can modulate brain activity and signaling (*Islam et al., 2021*; *Wrann et al., 2013*; *Young et al., 2019*; *Jo and Song, 2021*; *Qi et al., 2022*) through BDNF (*Wrann et al., 2013*) and BDNF promotes osteogenesis in human bone mesenchymal stem cells (*Liu et al., 2018*). Our data do not show significant expression of *Fndc5* in osteocytes. Studies from our group have found no expression of *Fndc5* in primary osteoblasts and primary osteocytes (transcriptome analysis with a raw count of 8–12), however both skeletal muscle (gastrocnemius) and C2C12 myotubes have high expression of *Fndc5* (transcriptome raw count of 512–1000, unpublished). As such, we postulate that the effect of irisin on osteocytes is not an autocrine effect, but rather due to irisin production by skeletal muscle.

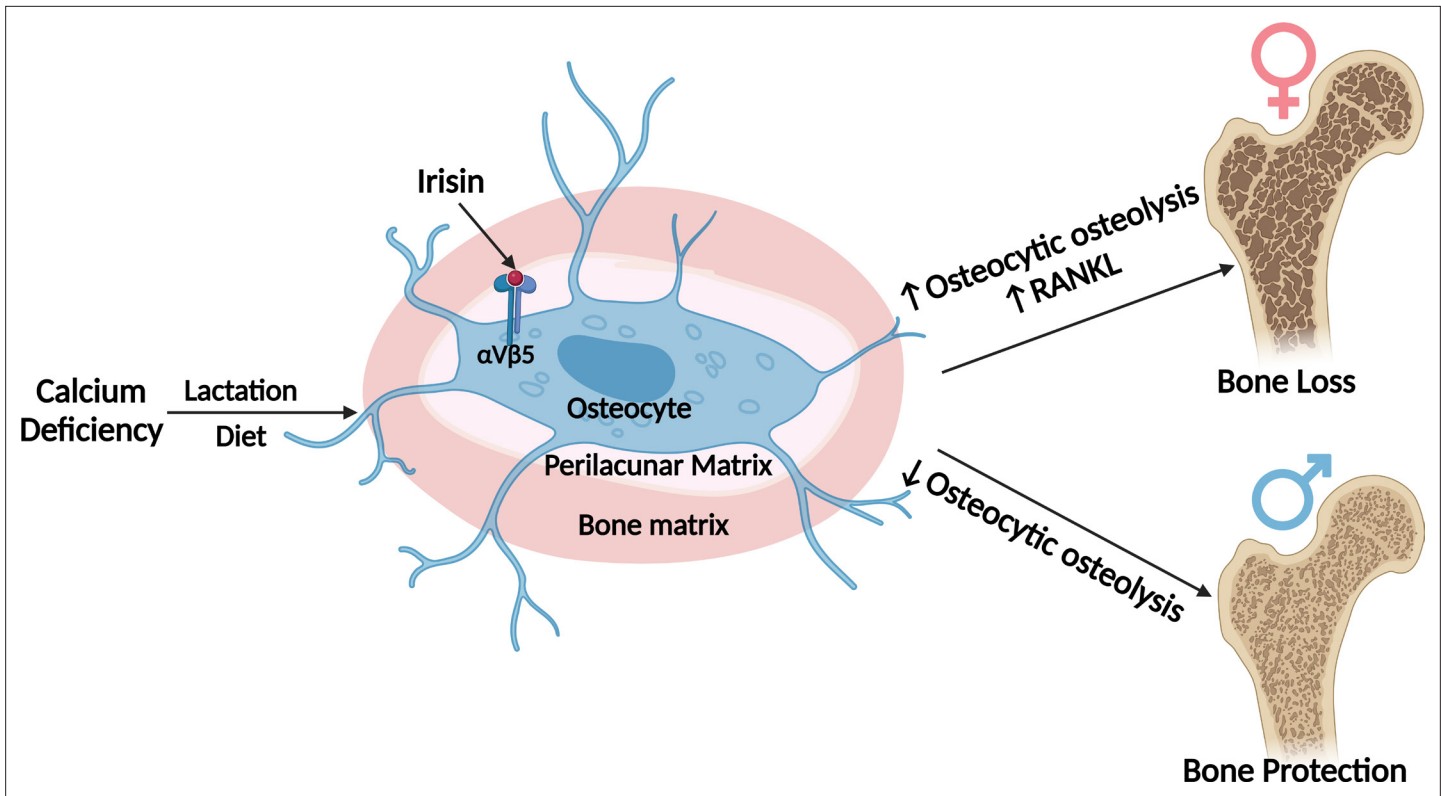

**Figure 8.** Graphical abstract (image was created using BioRender.com and published using a CC BY-NC-ND license with permission). No differences are observed in bone from Fndc5/irisin null female, whereas null male skeletons are larger but weaker compared to wildtype controls. With calcium deficiency, lactating female null mice are protected from bone loss due to osteocytic osteolysis, whereas male null mice on a low-calcium diet lose greater amounts of bone compared to their wildtype controls. The osteocyte transcriptomes show wildtype males have higher expression of the steroid, lipid, and fatty acid pathways which are lower in the null males, whereas the wildtype females have higher expression of genes regulating osteocytic osteolysis than null females. With calcium deficiency, female null osteocytes have lower while male null osteocytes have higher expression of osteocytic osteolysis genes compared to wildtype controls.

Irisin must bind to $\alpha V\beta 5$ integrins to function. Osteocytes express high levels of this receptor which was first discovered using the female MLO-Y4 osteocyte-like cell line (*Kim et al., 2018*). Integrins are usually stable in the cell membrane with a half-life of 12–24 hr (*Moreno-Layseca et al., 2019*). In our RNA sequencing data, we observed a stable expression of both *ITGAV* and *ITGB5*, encoding integrins $\alpha V$ and $\beta 5$ respectively, with no differences between either WT or null, male or female, calcium-replete or calcium-deficient mice. Recently it has been published that Hsp90$\alpha$ is necessary to facilitate irisin-$\alpha V\beta 5$ binding (*Mu et al., 2023*). *Hsp90a*, the gene encoding this heat shock protein, is very highly expressed in both WT and null male and female mice, with no significant regulation by diet. The high expression of Hsp90$\alpha$ in osteocytes may explain their significant and rapid responses to irisin (*Kim et al., 2018*).

In summary, during normal development and on a regular diet, FNDC5/irisin deletion has few if any effects on the female skeleton but a significant effect on the male skeleton resulting in more but weaker bone. However, with challenges, such as calcium deficiency, dramatic differences were observed. Our data suggest that irisin activates the osteocyte in females to initiate the removal of their perilacunar matrix and for bone resorption through osteoclast activation, presumably to provide calcium for reproduction purposes. In contrast, in males, irisin protects against osteocytic osteolysis and osteoclastic bone resorption under calcium-demanding conditions. This sex-specific effect may be due to the sexual dimorphism of the osteocyte transcriptome. The major findings of our work is summarized in (*Figure 8*). We have discovered a new novel function of irisin to ensure the survival of offspring and that irisin is essential for male but not female skeletal development. These findings

could have implications for understanding sex-dependent differences in bone diseases, such as osteoporosis, and lead to the development of sex-targeted therapies.

## Methods

### Animal experiments

All animal experiments were performed per procedures approved by the Institutional Animal Care and Use Committee (IACUC) of the Indiana University School of Medicine. Heterozygous C57Bl/6J FNDC5 KO mice were provided by Dr. Bruce Spiegelman at Harvard University and bred in our facility to obtain homozygous global FNDC5 KO and WT control mice. Genotype was determined using a PCR with primers targeting portions of exon 3 absent in KO (WT Forward: GCG GCT CGA GAG ATG AAG AA, WT Reverse: CAG CCC ACA ACA AGA AGT GC, KO Forward: GGA CTT CAA GTC CAA GGT CA, KO Reverse: CCT AAG CCC ACC CAA ATT AC). Mice were housed in a temperature-controlled (20–22°C) room on a 12 hr light/dark cycle with ad libitum food and water. Qualified veterinary staff and/or animal care technicians performed regular health check inspections.

For the lactation experiments, 4-month-old WT and FNDC5 global KO female mice were bred, delivered pups, and lactated for 2 weeks before sacrifice. Virgin WT and KO mice were used as controls. All animals were 4–5 months of age at the time of sacrifice and analysis. For all lactating mice, the litter size ranged from 8 to 11 pups (*Qing et al., 2012*).

For the low-calcium diet experiments, 4- to 5-month-old male and female WT and FNDC5 global KO mice were fed either a control diet (0.6% calcium, Teklad, TD.97191) or a low-calcium diet (0.01% calcium, 0.4% phosphorus, Teklad TD.95027) for 2 weeks. Food was replaced every 2 days. Distilled water was used in place of tap water to control calcium intake. On the day of sacrifice, blood was collected under anesthesia, and mice were euthanized for sample collection, processing, and analysis (*Qing et al., 2012*; *Jähn et al., 2017*).

### AAV8 injection

AAV8-irisin and AAV8-GFP constructs were obtained from Dr. Bruce Spiegelman at Harvard University. AAV8 Mouse ORF 1–140 (containing the N-terminal signal peptide and irisin) plus a five-amino acid linker plus a C-terminal flag tag was cloned into the pENN.AAV.CB7.CI.pm20d1flag.WPRE.rBG vector (Addgene plasmid no. 132682). AAV8-GFP (pENN.AAV.CB7.CI.eGFP.WPRE.rBG), used as control, was obtained from Addgene (105542), and packaged at the UPenn Vector Core to a titer of $2.10×10^{13}$GC per mL. FNDC5 KO male mice were placed under anesthesia and injected into the tail vein with either AAV8-irisin or AAV8-GFP control ($1×10^{10}$ GC per mouse) in 100 µL in PBS (*Islam et al., 2021*). One week after injection with either the control virus containing GFP or the virus coding for circulating irisin, the mice were placed on a low-calcium diet for 2 weeks before sacrifice.

### In vivo and ex vivo muscle contractility and electrophysiology measurement

In vivo plantarflexion torque was assessed 1 day before sacrifice (Scientific Inc, Canada) as described in *Pin et al., 2020*. Briefly, the mouse was placed under anesthesia and the left hind foot was affixed to the force transducer aligned with the tibia at 90°. The tibial nerve was stimulated using monopolar electrodes (Natus Neurology, Middleton, WI, USA). Maximum twitch torque was established by using a 0.2 ms square wave pulse. Peak plantarflexion torque was measured by using a stimulation of 0.2 ms delivered at 100 Hz stimulation frequency.

In vivo electrophysiological functions were assessed 1 day before sacrifice with the Sierra Summit 3–12 Channel EMG (Cadwell Laboratories Incorporated, Kennewick, WA, USA) as described in *Huot et al., 2022*. Briefly, peak-to-peak and baseline-to-peak compound muscle action potentials (CMAP) were measured using supramaximal stimulations of <10 mA continuous current for 0.1 ms duration, and peak-to-peak single motor unit (SMUP) potentials were measured using an incremental stimulation technique. Motor unit number estimation (MUNE) was measured using the equation: MUNE = CMAP amplitude/average SMUP.

Ex vivo muscle contractility was measured in the extensor digitorum longus (EDL) muscle as described in *Huot et al., 2021*. EDL was collected from the mouse and mounted between a force transducer, and then submerged in a stimulation bath. The muscles were forced to contract, and data

were collected using Dynamic Muscle Control/Data Acquisition (DMC) and Dynamic Muscle Control Data Analysis (DMA) programs (Aurora Scientific). The EDLs were weighed for normalization purposes.

## Body composition assessment by DXA

The right femurs from mice were dissected and cleaned of soft tissue, fixed in 4% paraformaldehyde (PFA) for 48 hr, and then transferred to 70% ethanol. Ex vivo dual-energy X-ray absorptiometry (DXA) measurements were obtained using a faxitron (Faxitron X-ray Corp, Wheeling, IL, USA) to measure BMD and BMC (*Essex et al., 2020*).

## Bone morphometry analysis by μCT

Right femurs were analyzed using a Skyscan 1176 micro-computed tomography (μCT) as described previously (*Pin et al., 2020*). Briefly, specimens were scanned at 55 kV, 145 μA, high resolution, 10.5 mm voxel, and 200 ms integration time. For cortical parameters, 3D images from a 1 mm region of interest of the mid-diaphysis were used to calculate total cortical bone area fraction (Ct. B. Ar/T. Ar%), cortical bone thickness (Ct. Th), marrow cavity area, periosteal perimeter (Ps. Pm), and endosteal perimeter (Es. Pm) according to ASBMR guidelines (*Bouxsein et al., 2010*). For trabecular parameters, 3D images reconstructed within the range of 0.5 mm from the most proximal metaphysis of tibiae were analyzed. Trabecular morphometry was performed by excluding the cortical bone from the endocortical borders using hand-drawn contours followed by thresholding and characterized by BV/TV, Tb. N, Tb. Th, Tb. Sp, and connectivity density (*Kitase et al., 2018*).

## TRAP staining

Tibiae were stripped of soft tissue, fixed in 4% PFA for 48 hr, decalcified in 10% EDTA for 3–4 weeks, and processed into paraffin as described previously followed by sectioning (5 μm) and staining for TRAP activity using the standard naphthol AS-BI phosphate post coupling method and counterstained with toluidine blue (*Pin et al., 2021*). Briefly, after equilibration in 0.2 M sodium acetate, 50 mM sodium tartrate, pH 5.0, for 20 min at room temperature (RT), sections were incubated at 37°C in the same buffer containing 0.5 mg/mL naphthol AS-MX phosphate (Sigma Chem. Co., St. Louis, MO, USA) and 1.1 mg/mL Fast Red Violet LB salt (Sigma) and counterstained in toluidine blue. Images were taken at 5× and 40× using an Olympus BX51 fluorescent microscope and Olympus cellSense Entry 1.2(Build 7533) imaging software. TRAP-positive osteocytes and osteoclasts 1.5 mm distal from the growth plate were quantified using Osteomeasure software (OsteoMetrics Inc) in a blind fashion. Toluidine blue-stained osteoblasts from the same sections were quantified 1.5 mm distal from the growth plate using the same software.

## Osteocyte lacunar area measurement by BSEM

Femurs were stripped of soft tissue and fixed in 4% PFA for 48 hr before proceeding to dehydration and embedding steps as previously described (*Qing et al., 2012*). Briefly, femurs were dehydrated in graded ethanol and placed into acetone. Subsequently, the femurs were immersed in infiltration solution made of 85% destabilized methyl methacrylate (MMA, Sigma), 15% dibutyl phthalate (Sigma), 1% PEG400 (Sigma), and 0.7% benzoyl peroxide (Polysciences, Inc, Warrington, PA, USA)/acetone until infiltration was complete. The femurs were then placed on pre-polymerized base layers, covered with freshly catalyzed MMA embedding solution (for 100 mL, 85 mL MMA, 14 mL dibutyl phthalate, 1 mL PEG400, 0.33 μL DMT, and 0.8 g BPO), and incubated under vacuum until the MMA was polymerized. The polymerized blocks were trimmed, sequentially polished to a completely smooth surface, and coated with gold using a sputter coater (Desk V, Denton Vacuum, NJ, USA). Then backscatter scanning electron microscopy (BSEM) (JEOL: JSM-7800F) was performed to image the osteocyte lacunae on the sectioned bone surface at ×450 magnification starting 2 mm distal from the growth plate. Six fields from the endosteal and periosteal sides of the cortical bone were taken as described previously (*Qing and Bonewald, 2009*). Using ImageJ (NIH), the images were thresholded for background removal, binarized, and the lacunar area from each sample quantitated.

## Mechanical testing using three-point bending

Mechanical testing was performed essentially as described in *Melville et al., 2015*. Briefly, the left femurs were stripped of soft tissue, wrapped in PBS-soaked gauze, and stored at –20°C until use.

Frozen femurs were brought to RT and mounted across the lower supports (8 mm span) of a three-point bending platen on a TestResources R100 small force testing machine. The samples were tested in monotonic bending to failure using a crosshead speed of 0.05 mm/s. Parameters related to whole bone strength were measured from force/displacement curves.

### Serum RANKL analysis

The levels of RANKL were measured in mouse centrifuged serum by using an ELISA kit (Bio-Techne Corporation, Minneapolis, MN, USA), according to the manufacturer's protocol.

### Serum PTH analysis

Serum was obtained from terminal cardiac puncture and serum PTH levels were determined using the MicroVue Bone Mouse PTH 1-84 ELISA assay (Quidel Corp., San Diego, CA, USA) according to the manufacturer's protocol.

### Calcium measurement

Plasma calcium levels were determined using the Pointe Scientific calcium Reagent kit (Pointe Scientific, Michigan, USA). Briefly, diluted serum (1:4 in $dH_2O$) was incubated with a working calcium color reagent for 1 min and the absorbance read at 575 nm using a spectrophotometer (BioTek Synergy HTX).

### Sample collection and processing for RNA sequencing

Bulk RNA sequencing was performed on osteocytes from the control and low-calcium diet, male and female, WT and KO mice. Osteocyte RNA was extracted from tibia and femur diaphyses after sequential digestion to remove surface cells including osteoclasts, osteoblasts, and lining cells as previously described (*Qing et al., 2012*; *Pin et al., 2020*). Briefly, soft tissue was removed from the bones, the epiphyses were cut off and bone marrow was removed by flushing with PBS. The remaining midshafts were incubated at 37°C with 0.2% type 1 collagenase (Sigma) for 30 min, followed by chelation/digestion in 0.53 mM EDTA/0.05% trypsin (Cellgro, Mediatech, Inc, Manassas, VA, USA) at 37°C for 30 min followed by a second collagenase digestion. After each step, the bone chips were rinsed with PBS and after the final step, flash-frozen in liquid nitrogen, and pulverized in liquid nitrogen, with Trizol reagent (QIAGEN, Carlsbad, CA, USA) added to the resulting bone powder. Total RNA was isolated with an RNA purification kit (QIAGEN miRNeasy mini kit) and DNase treatment to remove DNA contamination.

### Library preparation and RNA sequencing

Total RNA samples were first evaluated for their quantity and quality using Agilent TapeStation. All the samples used for the sequencing had a RIN of at least 5. 100 ng of total RNA were used for library preparation with the KAPA total RNA Hyperprep Kit (KK8581) (Roche). Each resulting uniquely dual-indexed library was quantified and quality accessed by Qubit and Agilent TapeStation. Multiple libraries were pooled in equal molarity. The pooled libraries were sequenced on an Illumina NovaSeq 6000 sequencer with the v1.5 reagent kit. 100 bp paired-end reads were generated.

### RNA sequencing data analysis

The sequencing reads were first quality-checked using FastQC (v0.11.5, Babraham Bioinformatics, Cambridge, UK) for quality control. The sequence data were then mapped to the mouse reference genome mm10 using the RNA sequencing aligner STAR (v2.7.10a) (*Dobin et al., 2013*) with the following parameter: '--outSAMmapqUnique60'. To evaluate the quality of the RNA sequencing data, the number of reads that fell into different annotated regions (exonic, intronic, splicing junction, intergenic, promoter, UTR, etc.) of the reference genome was assessed using bamutils (*Breese and Liu, 2013*). Uniquely mapped reads were used to quantify the gene-level expression employing featureCounts (subread v2.0.3) (*Liao et al., 2014*) with the following parameters: '-s 2 -Q 10'.

### Quality control of samples

During data quality control, one of the KO female control samples (sample 23) was found to have a similar proportion of reads on chromosome Y as in male mice and a very low expression of the gene

Xist, typically highly expressed in females (*Figure 5—figure supplement 1A and B*), therefore this sample was excluded from the analysis.

The WT female low-calcium diet samples (samples 16, 17, and 18) had low mapping percentages of 37%, 32%, and 61%, respectively. This may be due to bacterial contamination. The two possible methods to process these data are to filter all the possible contaminated reads before alignment or align the reads without filtering. However, filtering the possible contaminated reads before alignment may result in removing some reads from the mouse genome which is similar to the bacterial genome (causing lower gene expression). In contrast, using data without filtering may result in some genes having higher expression levels due to reads from the bacterial genome which are aligned to mice genes. We decided to perform a principal component analysis using data without filtering and found that the samples clearly clustered into four groups: control male mice, control female mice, low-calcium diet male mice, and low-calcium diet female mice (*Figure 5—figure supplement 1C*). Within each group, the separation of WT and KO mice is also clear. Due to contamination, samples 16 and 18 were slightly far apart from the others. However, contamination should not have a large global influence on the data as samples 16, 17, and 18 are close to the non-contaminated samples 5 and 6, also in the low-calcium diet female group. Additionally, we validated the data using quantitative polymerase chain reaction (qPCR) with selected genes.

## DEG analysis

The read counts matrix was imported to *Team RC, 2022*, and analyzed with DEseq2 (*Love et al., 2014*). Within DESeq2, read counts data were normalized with median of ratios, and DEGs were detected after independent filtering. In DEG analysis, we first detected DEGs between different groups. Significant genes were defined as genes with an unadjusted p-value less than 0.01 and absolute log2 fold-change larger than 1. Gene set enrichment analysis was applied on gene sets from *Gene Ontology Consortium, 2021*, using R package clusterProfiler (*Wu et al., 2021*). p-Value of less than 0.05 considered as significant for the gene ontology analysis. Several RNA sequencing and pathway figures were prepared with R packages ggplot2 (*Wickham, 2016*) and ComplexHeatmap (*Gu, 2022*). The data was deposited in NCBI GEO database (accession number GSE242445).

## Real-time qPCR

Total RNA was reverse-transcribed to cDNA using the Verso cDNA Kit (Thermo Fisher Scientific). Transcript levels were measured by real-time PCR (Light Cycler 96; Roche), taking advantage of the TaqMan and SYBR Gene Expression Assay System (Thermo Fisher Scientific). Expression levels for RANKL (*Tnfsf11*, Forward primer: CCG AGC TGG TGA AGA AAT TAG, Reverse: CCC AAA GTA CGT CGC ATC TTG), Cathepsin K (*Ctsk*, Primer Bank ID: Mm.PT.58.9655974, IDT), TRAP (*Acp5*, Mm.PT.58.5755766, IDT), and sclerostin (*Sost*, Mm00470479_m1, Applied Biosystems) were quantitated. Gene expression was normalized to β-2-microglobulin (*B2m*, Forward: ACA GTT CCA CCC GCC TCA CAT T, Reverse: TAG AAA GAC CAG TCC TTG CTG AAG) levels using the standard 2-ΔΔCt method.

## Statistical analysis

Data are expressed as individual data points. The statistical analysis was done by Prism 8.2 (GraphPad Software, San Diego, CA, USA) and R 4.3.0. When comparing three or more groups with two variables, a two-way analysis of variance (ANOVA) was used. To compare between two groups, the unpaired, two-tailed Student's t-test was used. Differences were considered significant at *p<0.05, **p<0.01, and ***p<0.001.

## Acknowledgements

We would like to thank the Center for Medical Genomics, the Small Animal Phenotypic Core, and the Histology and Histomorphometry Core at the Indiana Center for Musculoskeletal Health for help and advice with histological sample preparation. We would like to thank Dr. Yukiko Kitase, Dr. Eijiro Sakamoto, and Carrie Zhao for their help and advice with the experiments. This work was supported by NIH awards P01 AG039355 (to LFB).

# Additional information

## Funding

| Funder | Grant reference number | Author |
|---|---|---|
| NIH Office of the Director | PO1039355 | Lynda F Bonewald |

The funders had no role in study design, data collection and interpretation, or the decision to submit the work for publication.

## Author contributions

Anika Shimonty, Data curation, Formal analysis, Investigation, Writing - original draft, Writing – review and editing; Fabrizio Pin, Matthew Prideaux, Data curation, Formal analysis, Writing – review and editing; Gang Peng, Data curation, Formal analysis, Visualization, Writing – review and editing; Joshua Huot, Data curation, Formal analysis, Methodology, Writing – review and editing; Hyeonwoo Kim, Conceptualization, Resources, Writing – review and editing; Clifford J Rosen, Conceptualization, Project administration, Writing – review and editing; Bruce M Spiegelman, Conceptualization, Resources, Project administration, Writing – review and editing; Lynda F Bonewald, Conceptualization, Resources, Supervision, Writing - original draft, Project administration, Writing – review and editing

## Author ORCIDs

Anika Shimonty ⬡ http://orcid.org/0009-0005-2954-2991
Matthew Prideaux ⬡ http://orcid.org/0000-0001-9211-9698
Lynda F Bonewald ⬡ http://orcid.org/0000-0002-5536-9943

## Ethics

This study was performed in strict accordance with the recommendations in the Guide for the Care and Use of Laboratory Animals of the National Institutes of Health. All of the animals were handled according to approved institutional animal care and use committee (IACUC) protocols (#20083 for lactation and low calcium diet experiments and #22051 for muscle function studies) of Indiana University. No surgeries were performed on these animals.

Reviewer #1 (Public Review): https://doi.org/10.7554/eLife.92263.3.sa1
Reviewer #2 (Public Review): https://doi.org/10.7554/eLife.92263.3.sa2
Reviewer #3 (Public Review): https://doi.org/10.7554/eLife.92263.3.sa3
Author response https://doi.org/10.7554/eLife.92263.3.sa4

# Additional files

## Supplementary files

• Supplementary file 1. FNDC5 knockout (KO) mice femurs are partially resistant to lactation-induced bone loss. Femoral cortical and trabecular bone parameters of wildtype (WT) and FNDC5 KO female virgin and lactation mice. n = 5–8/group. Data presented as mean ± standard deviation. a=Significant compared to WT control, b=significant compared to KO control, c=significant compared to WT low-calcium diet, two-way analysis of variance (ANOVA), significance <0.05, n = 8/group. Percentage change in different bone and serum parameters in WT and FNDC5 KO female mice with lactation. *=p<0.05 compared to WT.

• Supplementary file 2. Wildtype (WT) and FNDC5 knockout (KO) female and male mice bone responds differently to a low-calcium diet. Femoral bone mineral density (BMD), bone mineral content (BMC), cortical and trabecular bone parameters, and mechanical properties of 4- to 5-month-old WT and KO female and male mice under a normal diet or a 2-week low-calcium diet. n = 5/group. Data presented as mean ± standard deviation. a=significant compared to WT control, b=significant compared to KO control, c=significant compared to WT low-calcium diet, two-way analysis of variance (ANOVA), significance <0.05, n = 4–5/group.

• Supplementary file 3. Differentially expressed genes in WT F vs WT M and KO F vs KO M. Genes that are significantly differentially expressed in female FNDC5 KO mice compared to female WT mice as well as genes that are significantly differentially expressed in male FNDC5 KO mice

compared to male WT mice are listed.

• Supplementary file 4. Female KO lc vs WT lc and male KO lc vs WT lc genes. Genes that are significantly differentially expressed in low-calcium diet-fed female FNDC5 KO mice compared to low-calcium diet-fed female WT mice as well as genes that are significantly differentially expressed in low-calcium diet-fed male FNDC5 KO mice compared to low-calcium diet-fed male WT mice are listed.

• MDAR checklist

## Data availability

The osteocyte transcriptome sequencing data have been deposited in GEO under accession number GSE242445.

The following dataset was generated:

| Author(s) | Year | Dataset title | Dataset URL | Database and Identifier |
|---|---|---|---|---|
| Shimonty A, Pin F, Prideaux M, Peng G, Huot JR, Kim H, Rosen CJ, Spiegelman BM, Bonewald LF | 2024 | Deletion of FNDC5/Irisin modifies murine osteocyte function in a sex-specific manner | https://www.ncbi.nlm.nih.gov/geo/query/acc.cgi?acc=GSE242445 | NCBI Gene Expression Omnibus, GSE242445 |

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
