## [Editor Report · eLife assessment]

The study presents **valuable** findings on sexually dimorphic patterns of osteocytic transcriptomes and low calcium diet-induced osteocytic osteolysis in FNDC5-deficient mice. The authors present **solid** evidence for sex-specific changes in osteocyte morphology and gene expression under a calcium-demanding setting in this particular strain of mice, although the protective role of FNDC5-deficiency in lactation and low-calcium diet in female mice remains unclear due to lack of mechanistic studies. The study also lacks evidence that irisin, a proteolytically cleaved product of FNDC5, is responsible for the observed phenotypes, as irisin was not directly measured.

---

## [Referee Report · Reviewer #1 (Public Review)]

In this manuscript, Shimonty and colleagues study the effects of FNDC5/irisin deletion on osteocytes in a sex-specific manner using models of lactation induced bone loss and bone loss due to low calcium diet (LCD). Consistent with the previous findings of Kim et al. (2018), the authors report 'protective' effects of irisin deficiency in lactating female FNDC5-null mice due to reduced osteocytic osteolysis. Interestingly, FNDC5 null mice show distinct changes when placed on LCD, with mutant females showing some protection from hyperparathyroidism-induced bone loss, while mutant males (which have more cortical bone at baseline) show increased LCD-induced bone loss. Furthermore, new insights into irisin's role in osteocytes regarding cellular energetic metabolism were provided by sex and gene-dependent transcriptomic datasets. Strengths of the well-written manuscript include clear description of sex-dependent effects, strong transcriptomic datasets, and focus on cortical bone changes using microCT, histomorphometry, BSEM, and serum analysis. Despite these strengths, important weaknesses are noted (below) which could be addressed to improve the impact of the work for a broad audience.

Major comments:

(1) Overall, the magnitude of the effect size due to FNDC5 deficiency in both male and female mice is rather modest at the level of bone mass. Looking at the data from a qualitative perspective, it is clear that knockout females still lose bone during lactation and on the low calcium diet (LCD). It is difficult to assess the physiologic consequence of the modest quantitative 'protection' seen in FNDC5 mutants since the mutants still show clear and robust effects of lactation and LCD on all parameters measured. Similarly, the magnitude of the 'increased' cortical bone loss in FNDC5 mutant males is also modest, and perhaps could be related to the fact that these mice are starting with slightly more cortical bone. Since the authors do not provide a convincing molecular explanation for why FNDC5 deficiency causes these somewhat subtle changes, I would like to offer a suggestion for the authors to consider (below, point #2) which might de-emphasize the focus of the manuscript on FNDC5. If the authors chose not to follow this suggestion, the manuscript could be strengthened by addressing the consequences of the modest changes observed in WT versus FNDC5 KO mice. I understand that the effects of FNDC5 are more obvious at the level of osteocyte morphology, and it is reasonable to emphasize these findings here.

(2) The bone RNA-seq findings reported in Figures 4-6 are quite interesting. Although Youlten et al previously reported that the osteocyte transcriptome is sex-dependent, the work here certainly advances that notion to a considerable degree, and likely will be of high interest to investigators studying skeletal biology and sexual dimorphism in general. To this end, one direction for the authors to consider might be to refocus their manuscript towards sexually-dimorphic gene expression patterns in osteocytes and the different effects of LCD on male versus female mice. This would allow the authors to better emphasize these major findings, and then to use FNDC5 deficiency as an illustrative example of how sexually-dimorphic osteocytic gene expression patterns might be affected by deletion of an osteocyte-acting endocrine factor. Ideally, the authors would confirm RNA-seq data comparing male versus female mice in osteocytes using in situ hybridization or immunostaining. Of course, this point is only a suggestion for the authors to consider.

(3) It would be appreciated if the authors could provide additional serum parameters (if possible) to clarify incomplete data in both lactation and low-calcium diet models: RANKL/OPG ratio, Ctx, PTHrP, and 1,25-dihydroxyvitamin D levels. I understand that this may not be possible due to lack of available material.

---

## [Referee Report · Reviewer #2 (Public Review)]

Summary:

The goal of this study was to examine the role of FNDC5 in the response of the murine skeleton to either lactation or a calcium-deficient diet. The authors find that female FNDC5 KO mice are somewhat protected from the bone loss and osteocyte lacunar enlargement caused by either lactation or a calcium-deficient diet. In contrast, male FNDC5 KO mice lose more bone and have a greater enlargement of osteocyte lacunae than their wild type controls. Based on these results, the authors conclude that in males irisin protects bone from calcium deficiency but that in females it promotes calcium removal from bone for lactation.

While some of the conclusions of this study are supported by the results, it is not clear that the modest effects of FNDC5 deletion have an impact on calcium homeostasis or milk production.

Specific comments.

(1) The authors sometimes refer to FNDC5 and other times to irisin when describing causes for a particular outcome. Because irisin was not measured in any of the experiments, the authors should not conclude that lack of irisin is responsible. Along these lines, is there any evidence that either lactation or a calcium-deficient diet increases production of irisin in mice?

(2) The results of the irisin-rescue experiment shown in figure 2G cannot be appropriately interpreted without normal diet controls. In addition, some evidence that the AAV8-irisin virus actually increased irisin levels in the mice would strengthen the conclusion.

(3) There is insufficient evidence to support the idea that the effect of FNDC5 on bone resorption and osteocytic osteolysis is important for the transfer of calcium from bone to milk. Previous studies by others have shown that bone resorption is not required to maintain milk or serum calcium when dietary calcium is sufficient but is critical if dietary calcium is low (Endo. 156:2762-73, 2015). To support the conclusions of the current study, it would be necessary to determine whether FNDC5 is required to maintain calcium levels when lactating mice lack sufficient dietary calcium.

(4) The amount of cortical bone loss due to lactation is very similar in both WT and FNDC5 KO mice. The results of the statistical analysis of the data presented in figure 1B are surprising given the very similar effect size of lactation. The key result from the 2-way ANOVA is whether there is an effect of genotype on the effect size of lactation (genotype-lactation interaction). The interaction terms were not provided. Similar concerns are noted for the results shown in figure 1G and H.

(5) It is not clear what justifies the term 'primed' or 'activated' for resorption. Is there evidence that a certain level of TRAP expression lowers the threshold for osteocytic osteolysis in response to a stimulus?

---

## [Referee Report · Reviewer #3 (Public Review)]

Summary: Irisin has previously been demonstrated to be a muscle-secreted factor that affects skeletal homeostasis. Through the use of different experimental approaches, such as genetic knockout models, recombinant Irisin treatment, or different cell lines, the role of Irisin on skeletal homeostasis has been revealed to be more complex than previously thought and this warrants further examination of its role. Therefore, the current study sought to rigorously examine the effects of global Irisin knockout (KO) in male and female mouse bone. Authors demonstrated that in calcium-demanding settings, such as lactation or low-calcium diet, female Irisin KO mice lose less bone compared to wildtype (WT) female mice. Interestingly male Irisin KO mice exhibited worse skeletal deterioration compared to WT male mice when fed low-calcium diet. When examined for transcriptomic profiles of osteocyte-enriched cortical bone, authors found that Irisin KO altered the expression of osteocytic osteolysis genes as well as steroid and fatty acid metabolism genes in males but not in females. These data support authors' conclusion that Irisin regulates skeletal homeostasis in a sex-dependent manner.

Strengths:

The major strength of the study is rigorous examination of the effects of Irisin deletion in the settings of skeletal maturity and increased calcium demands in female and male mice. Since many of the common musculoskeletal disorders are dependent on sex, examining both sexes in the preclinical setting is crucial. Had the investigators only examined females or males in this study, the conclusion from each sex would have contradicted each other regarding the role of Irisin on bone. Also, the approaches are thorough and comprehensive that assess the functional (mechanical testing), morphological (microCT, BSEM, and histology), and cellular (RNA-seq) properties of bone. Transcriptomic data deposited to NCBI GEO data repository will be a valuable resource to musculoskeletal researchers who aim to further assess the affects of Irisin on skeleton.

Weaknesses:

One of the weaknesses of this study is a lack of detailed mechanistic analysis of why Irisin has sex-dependent role on skeletal homeostasis. However, the osteocyte transcriptome comparisons between LC females vs. LC males lay a foundation for such future mechanistic studies.

Another weakness is authors did not present data that convincingly demonstrate that Irisin secretion is altered in the skeletal muscle between female vs. male WT mice in response to calcium restriction. The supplement skeletal muscle data only present functional and electrophysiological outcomes. Since Itgav or Itgb5 were not different in any of the experimental groups, it is assumed that the changes in the level of Irisin is responsible for the phenotypes observed in WT mice. Assessing Irisin expression will further strengthen the conclusion based on observing skeletal changes that occur in Irisin KO male and female mice.

---

## [Author Response]

The following is the authors’ response to the original reviews.

Overall, the magnitude of the effect size due to FNDC5 deficiency in both male and female mice is rather modest. Looking at the data from a qualitative perspective, it is clear that knockout females still lose bone during lactation and on the low calcium diet (LCD). It is difficult to assess the physiologic consequence of the modest quantitative 'protection' seen in FNDC5 mutants since the mutants still show clear and robust effects of lactation and LCD on all parameters measured. Similarly, the magnitude of the 'increased' cortical bone loss in FNDC5 mutant males is also modest and perhaps could be related to the fact that these mice are starting with slightly more cortical bone. Since the authors do not provide a convincing molecular explanation for why FNDC5 deficiency causes these somewhat subtle changes, I would like to offer a suggestion for the authors to consider (below, point #2) which might de-emphasize the focus of the manuscript on FNDC5. If the authors chose not to follow this suggestion, the manuscript could be strengthened by addressing the consequences of the modest changes observed in WT versus FNDC5 KO mice.

Response: We agree that the magnitude of the effect size due to FNDC5 deficiency is modest with regards to the quantitative cortical bone parameters. However, if one examines the changes in osteocyte lacunar size and the mechanical properties of these bones, the differences are greater. As shown in Figure 3 E, the lacunar area of the WT females on a low calcium diet increases by over 30% and the KO by less than 20%, while in the males it is approximately 38% in WT compared to 46% in KO mice. According to Sims and Buenzli (PMID: 25708054) a potential total loss of ~16,000 mm3 (16 mL) of bone occurs through lactation in the human skeleton. This was based on our measurements in lactation-induced murine osteocytic osteolysis (Qing et al PMID: 22308018). They used our 2D section of tibiae from lactating mice showing an increase in lacunar size from 38 to 46 um2. In that paper we also showed that canalicular width is increased with lactation. Therefore, this would suggest a dramatic decrease in intracortical porosity due to the osteocyte lacunocanalicular system in female KO on a low calcium diet compared to WT females and a dramatic increase in KO males compared to WT males. Also, PTH was higher in the serum of female WT compared to female KO mice on a low calcium diet, the opposite for males in order to maintain normal calcium levels (See Table 1). Based on this data, using the FNDC5 null animals, we would speculate that the product of FNDC5, irisin, is having a highly significant effect on the ultrastructure of bone in both males and females challenged with a low calcium diet.

(2) The bone RNA-seq findings reported in Figures 4-6 are quite interesting. Although Youlten et al previously reported that the osteocyte transcriptome is sex-dependent, the work here certainly advances that notion to a considerable degree and likely will be of high interest to investigators studying skeletal biology and sexual dimorphism in general. To this end, one direction for the authors to consider might be to refocus their manuscript toward sexually-dimorphic gene expression patterns in osteocytes and the different effects of LCD on male versus female mice. This would allow the authors to better emphasize these major findings, and to then use FNDC5 deficiency as an illustrative example of how sexually-dimorphic osteocytic gene expression patterns might be affected by deletion of an osteocyte-acting endocrine factor. Ideally, the authors would confirm RNA-seq data comparing male versus female mice in osteocytes using in situ hybridization or immunostaining.

Response: Thank you for this suggestion. We have compared the different effects of LCD on male versus female mice in our revised version and have added a figure containing this information.

(3) Along the lines of point #2 (above), the presentation of the RNA-seq studies in Figures 4-6 is somewhat confusing in that the volcano plot titles seem to be reversed. For example, Figure 4A is titled "WT M: WT F", but the genes in the upper right quadrant appear to be up-regulated in female cortical bone RNA samples. Should this plot instead be titled "WT F: WT M"? If so, then all other volcano plots should be re-titled as well.

Response: We have now insured that the plots are appropriately labeled.

(4) Have the authors compared male versus female transcriptomes of LCD mice?

Response: We have now compared the male vs female transcriptomes of LCD mice and added an additional figure.

(5) It would be appreciated if the authors could provide additional serum parameters (if possible) to clarify incomplete data in both lactation and low-calcium diet models: RANKL/OPG ratio, Ctx, PTHrP, and 1,25-dihydroxyvitamin D levels.

Response: It is not possible to quantitate each of these as the serum has been exhausted. We have checked the RANKL/OPG ratio in the RNA seq and qPCR data using osteocyte enriched bone chips and found no difference.

(6) Lastly, the data that overexpressing irisin improved bone properties in Fig 2G was somewhat confusing. Based on Kim et al.'s (2018) work, irisin injection increased sclerostin gene expression and serum levels, thus reducing bone formation. Were sclerostin levels affected by irisin overexpression in this study? Was irisin's role in modulating sclerostin levels attenuated with additional calcium deficiency?

Response: We have not observed any differences in the osteocyte Sost mRNA expression between WT and KO normal and low-calcium-diet male and female mice in our RNAseq and qPCR data. As such, we did not check the Sost levels for the 2G experiment.

**Reviewer #2 (Public Review):**
Summary:The goal of this study was to examine the role of FNDC5 in the response of the murine skeleton to either lactation or a calcium-deficient diet. The authors find that female FNDC5 KO mice are somewhat protected from bone loss and osteocyte lacunar enlargement caused by either lactation or a calcium-deficient diet. In contrast, male FNDC5 KO mice lose more bone and have a greater enlargement of osteocyte lacunae than their wild-type controls. Based on these results, the authors conclude that in males irisin protects bone from calcium deficiency but that in females it promotes calcium removal from bone for lactation.While some of the conclusions of this study are supported by the results, it is not clear that the modest effects of FNDC5 deletion have an impact on calcium homeostasis or milk production.Specific comments:(1) The authors sometimes refer to FNDC5 and other times to irisin when describing causes for a particular outcome. Because irisin was not measured in any of the experiments, the authors should not conclude that lack of irisin is responsible. Along these lines, is there any evidence that either lactation or a calcium-deficient diet increases the production of irisin in mice?

therefore we have extrapolated that the observed effects are due to a lack of circulating irisin. However, this does not rule out that Fndc5 itself could have a function, but this would have to be most likely in muscle and not in the osteocyte as we do not detect significant levels of irisin in either primary osteoblasts nor primary osteocytes compared to muscle and C2C12 cells. As such, we concluded that the phenotypical differences we saw in our experiments are due to a lack of irisin. We now address the reviewer’s point in the discussion. The measurement of irisin in the circulation with lactation or with low calcium diet of normal mice has not been performed.

(2) The results of the irisin-rescue experiment shown in figure 2G cannot be appropriately interpreted without normal diet controls. In addition, some evidence that the AAV8-irisin virus actually increased irisin levels in the mice would strengthen the conclusion.

Response: We do not have the normal diet controls at this time. We have quantitate tagged irisin in other AAV experiments and found highly significant expression

(3) There is insufficient evidence to support the idea that the effect of FNDC5 on bone resorption and osteocytic osteolysis is important for the transfer of calcium from bone to milk. Previous studies by others have shown that bone resorption is not required to maintain milk or serum calcium when dietary calcium is sufficient but is critical if dietary calcium is low (Endo. 156:2762-73, 2015). To support the conclusions of the current study, it would be necessary to determine whether FNDC5 is required to maintain calcium levels when lactating mice lack sufficient dietary calcium.

Response: We agree that it would be important to measure calcium levels in the milk to test the hypothesis that FNDC5 is important to maintain calcium levels in milk. However, as the calcium levels are normal in the serum, we are assuming they are normal in milk. This would require future experiments.

(4) The amount of cortical bone loss due to lactation is very similar in both WT and FNDC5 KO mice. The results of the statistical analysis of the data presented in figure 1B are surprising given the very similar effect size of lactation. The key result from the 2-way ANOVA is whether there is an effect of genotype on the effect size of lactation (genotype-lactation interaction). The interaction terms were not provided. Similar concerns are noted for the results shown in figure 1G and H.

Response: We agree, thanks. We will now add the interaction terms in the figure legends.

(5) It is not clear what justifies the term 'primed' or 'activated' for resorption. Is there evidence that a certain level of TRAP expression lowers the threshold for osteocytic osteolysis in response to a stimulus?

Response: The number of TRAP positive osteocytes in female KO mice are lower than in female WT. The number of TRAP positive osteocytes are lower in WT males compared to WT females. We propose that irisin plays a role in the number of TRAP positive osteocytes in normal, WT females by readying or preparing these cells to rapidly respond to low calcium. We will use the term ‘primed’ and will not use the term ‘activated’. We are open to any terminology or description as to why this is observed and what irisin could be doing to the osteocyte.

**Reviewer #3 (Public Review):**
Summary:Irisin has previously been demonstrated to be a muscle-secreted factor that affects skeletal homeostasis. Through the use of different experimental approaches, such as genetic knockout models, recombinant Irisin treatment, or different cell lines, the role of Irisin on skeletal homeostasis has been revealed to be more complex than previously thought and this warrants further examination of its role. Therefore, the current study sought to rigorously examine the effects of global Irisin knockout (KO) in male and female mouse bone. Authors demonstrated that in calcium-demanding settings, such as lactation or low-calcium diet, female Irisin KO mice lose less bone compared to wild-type (WT) female mice. Interestingly male Irisin KO mice exhibited worse skeletal deterioration compared to WT male mice when fed a low-calcium diet. When examined for transcriptomic profiles of osteocyte-enriched cortical bone, authors found that Irisin KO altered the expression of osteocytic osteolysis genes as well as steroid and fatty acid metabolism genes in males but not in females. These data support the authors' conclusion that Irisin regulates skeletal homeostasis in sex-dependent manner.Strengths:The major strength of the study is the rigorous examination of the effects of Irisin deletion in the settings of skeletal maturity and increased calcium demands in female and male mice. Since many of the common musculoskeletal disorders are dependent on sex, examining both sexes in the preclinical setting is crucial. Had the investigators only examined females or males in this study, the conclusions from each sex would have contradicted each other regarding the role of Irisin on bone. Also, the approaches are thorough and comprehensive that assess the functional (mechanical testing), morphological (microCT, BSEM, and histology), and cellular (RNA-seq) properties of bone.Weaknesses: One of the weaknesses of this study is a lack of detailed mechanistic analysis of why Irisin has a sex-dependent role on skeletal homeostasis. This absence is particularly notable in the osteocyte transcriptomic results where such data could have been used to further probe potential candidate pathways between LC females vs. LC males.

Response: Our future studies will focus on understanding the molecular mechanism behind the sex-dependent effects of irisin. Our RNA seq data shows a significant difference in the lipid, steroid, and fat metabolism pathways between male and female mice, as well as between WT and KO mice. Future studies will focus on these pathways.

Another weakness is authors did not present data that convincingly demonstrate that Irisin secretion is altered in the skeletal muscle between female vs. male WT mice in response to calcium restriction. The supplement skeletal muscle data only present functional and electrophysiolgical outcomes. Since Itgav or Itgb5 were not different in any of the experimental groups, it is assumed that the changes in the level of Irisin is responsible for the phenotypes observed in WT mice. Assessing Irisin expression will further strengthen the conclusion based on observing skeletal changes that occur in Irisin KO male and female mice.Response: The problem is that the commercial assays for irisin are not dependable, and results can differ widely across and beyond the physiologic range of 1-10 ng/ml. In part this is due to the nature of the polyclonal antibodies used and the resultant cross reactivity with other proteins. It was shown in Islam et al, 2021 (Nature Metabolism) that the commercial ELISAs were completely unreliable in mice and the only reliable method of measuring circulating irisin is mass spectrometry.
**Reviewer #1 (Recommendations For The Authors):**
Minor comments:(1) Were there any low calcium diet food intake or body weight alterations between littermates and FDNC5 KO mice?

Response: Yes, and we can now include the body weight data and the food intake data in the supplement. We do not observe any significant difference between the groups.

(2) In Fig 1, ideally the authors would provide the osteocyte lacunar density along with the lacunar area.

Response: We do not observe any difference in osteocyte density in any of the groups. There is not sufficient time within 2 weeks to see a change in osteocyte density because there is no new bone formation.

(3) What is the author's comment on the involvement of irisin on TGF-B signaling since the authors observed peri lacunar remodeling in FDNC5 KO mice? Authors should also include this in the discussion section regarding the Irisin-TGF-B signaling in terms of observed increased matrix-related signals.

Response: Perilacunar modeling is the removal followed by the replacement of the perilacunar and pericanilucular matrix as occurs with lactation (Qing et al 2012). Osteocytic osteolysis is the first half of that process where the matrix is removed. Alliston and colleagues generated transgenic mice with reduced expression of the TGFb Type II receptor in mice by using the Dmp1-Cre (PMID: 32282961). They clearly found a significant difference in bone parameters, the appearance of the osteocyte lacunocanalicular network, and markers of the osteocyte perilacunar remodeling between the sexes, however they did not compare the lacunar remodeling process in males as compared to females. The females were subjected to lactation and were found to be resistant to osteocytic osteolysis. To compare males and females, they would have had to challenge both sexes to a high calcium demanding condition such as low calcium diet as performed in the current study. Their study does suggest that TGFβ is involved in the osteocytic osteolysis that occurs with lactation. However, as the null males showed an abnormal lacunocanlicular network compared to wildtype males, this does not necessarily indicate a defect in perilacunar remodeling. It is more likely that the defect occurred during bone formation when osteoblasts were differentiating into osteocytes. Therefore, we will reference this paper regarding the role of TGFβ in osteocytic osteolysis in females with lactation but not in the comparison of males to females. We have examined the normalized expression of TGFβ1, 2, and 3 in the present study and found no significant differences in TGFβ1 or 2 in any of the groups, but did find significantly higher expression of TGFβ3 in females compared to males for WT (fdr < 0.05), LCD WT (fdr < 0.05), and Control KO (p value < 0.01). Perhaps this isoform is playing a major role in osteocytic osteolysis that occurs with lactation.

(4) Did the authors compare the transcriptomic dataset between lactated female WT vs. KO groups? Or were the RNA-seq studies only performed on LCD study samples?

Response: We have examined RNA sequence on the LCD study samples, and not in the lactating females.

**Reviewer #2 (Recommendations For The Authors):**
Line 401 on page 14 states that the sexes respond differently to calcium deficiency. Lacunar area increases in both sexes, so the response is very similar. What appears to be different between the sexes is the role of FNDC5 in this process.

Response: Female WT mice have higher osteocyte lacunar area at baseline with normal diet compared to WT males. With the low calcium diet, lacunar area increases in both sexes, with female WTs having a greater increase. We agree that what appears to be different between the sexes is the role of FNDC5 when challenged with high calcium demand.

**Reviewer #3 (Recommendations For The Authors):**
The authors state in the abstract and discussion that 'We propose Irisin ensures the survival of offspring by targeting the osteocytes...'. However, this appears to be over interpretation of their findings as they have not assessed the number of offspring surviving to weaning or their growth rate between WT and KO breeders.

Response: That was a proposal and we agree that it could be an over interpretation. However we would like to keep this as a speculation that could be tested in future studies.

Figures 1 and 2 should include cortical Total Area (and maybe Marrow Cavity data from Supp as well). These data will help readers to assess whether the thinning of the cortex is driven by impaired periosteal expansion or accelerated endosteal resorption (or both). Marrow cavity area data seem to suggest increased endosteal resorption (Supp. Table 2), but unclear if periosteal expansion is altered.

Response: The data are included in the supplementary tables. We do not observe any difference in the periosteal area between the groups.

To further support the author's statement that male KO mice exhibit different material properties of bone compared to WT mice, estimated elastic modulus should be calculated from the stiffness data (see https://doi.org/10.1002/jbmr.2539).

Response: We looked at the elastic modulus and it requires a stress strain curve instead of the force displacement we used in our calculations, therefore we were not able to get the estimated elastic modulus from the raw data we have.

In Figure 3 there is no legend indicating females or males. Based on the data and results texts it is assumed that red is Female and blue is Male. However, please confirm in the figure legend.

Response: This is now added in the figure legends.

Transcriptomic data should be deposited to NCBI GEO data repository. Also, please indicate whether cutoff p-value for DEG analysis was adjusted or not.

Response: We have submitted our data to the GEO data repository: GSE242445. Significant genes were defined as genes with p-value less than 0.01 and absolute log2 fold change larger than 1. The p-value is not adjusted. This information is now added.

The statistical analysis section indicates that a two-way repeated-measure ANOVA was used. However, the data presented in the study are from independent groups, in which case repeated-measure statistical approaches should not be used. Please clarify the statistical tests that were used.

Response: We now use regular ANOVA instead of repeated-measure ANOVA. Repeated-measure ANOVA is used for paired tests. The data remain significant.

In summary, we thank the reviewers for their very useful and thoughtful suggestions for improving our manuscript.